



# A Fundamental climate data record of SMMR, SSM/I, and SSMIS brightness temperatures

Karsten Fennig[1], Marc Schröder[1], Axel Andersson[2], and Rainer Hollmann[1]

[1]Deutscher Wetterdienst, Offenbach, Germany
[2]Deutscher Wetterdienst, Hamburg, Germany

**Correspondence:** Karsten Fennig (karsten.fennig@dwd.de)

**Abstract.** The Fundamental Climate Data Record (FCDR) of Microwave Imager Radiances from the Satellite Application Facility on Climate Monitoring (CM SAF) comprises inter-calibrated and homogenised brightness temperatures from the Scanning Multichannel Microwave Radiometer (SMMR), the Special Sensor Microwave/Imager (SSM/I) and the Special Sensor Microwave Imager/Sounder SSMIS radiometers. It covers the time period from October 1978 to December 2015 including all available data from the SMMR radiometer aboard Nimbus-7 and all SSM/I and SSMIS radiometers aboard the Defence Meteorological Satellite Program (DMSP) platforms. SMMR, SSM/I and SSMIS data are used for a variety of applications, such as analyses of the hydrological cycle, remote sensing of sea ice or as input into reanalysis projects. The improved homogenisation and inter-calibration procedure ensures the long term stability of the FCDR for climate related applications. All available raw data records from different sources have been reprocessed to a common standard, starting with the calibration of the raw Earth counts, to ensure a completely homogenised data record. The data processing accounts for several known issues with the instruments and corrects calibration anomalies due to along-scan inhomogeneity, moonlight intrusions, sunlight intrusions, and emissive reflector. Corrections for SMMR are limited because the SMMR raw data records were not available. Furthermore, the inter-calibration model incorporates a scene dependent inter-satellite bias correction and a non-linearity correction to the instrument calibration. The data files contain all available original sensor data (SMMR: Pathfinder Level 1b) and meta-data to provide a completely traceable climate data record. Inter-calibration and Earth incidence angle normalisation offsets are available as additional layers within the data files in order to keep this information transparent to the users. The data record is complemented with noise equivalent temperatures (Ne$\Delta$T), quality flags, surface types, and Earth incidence angles. The FCDR together with its full documentation, including evaluation results, is freely available at: doi:10.5676/EUM_SAF_CM/FCDR_MWI/V003 (Fennig et al., 2017).

## 1 Introduction

Data from space-borne microwave imagers and sounders such as the Scanning Multichannel Microwave Radiometer (SMMR), Special Sensor Microwave/Imager (SSM/I) and the Special Sensor Microwave Imager/Sounder (SSMIS) are used for a variety of applications, such as analyses of the hydrological cycle (precipitation and evaporation) and related atmospheric and surface parameters (Andersson et al., 2011), as well as remote sensing of sea ice (Lavergne et al., 2019), soil moisture (Dorigo et al.,





2017), and land surface temperatures (Prigent et al., 2016). Carefully calibrated and homogenised radiance data records are a fundamental prerequisite for climate analysis, climate monitoring and reanalysis (Dee et al., 2011; Hersbach et al., 2018). Several National Meteorological and Hydrological Service (NMHS) centres assimilate microwave radiances directly. Forecast and reanalysis can thus benefit from a Fundamental Climate Data Record (FCDR) of brightness temperatures ($T_B$) (Poli et al., 2013). An FCDR is a well-characterised, long term data record, compiled from a series of similar, quality-controlled

and calibrated series of similar instruments. Each individual instrument and its successor must have sufficient calibration and overlap to enable a successful inter-calibration of the measurements from all instruments. This way the generation of products that are accurate and stable to support climate analysis is allowed (GCOS-200, 2017). FCDRs are a prerequisite for the generation of Thematic Climate Data Records (TCDRs). Highest possible TCDR quality can only be achieved by harmonising the satellite observations in radiance space, in turn increasing the product's value for users.

The aim of the CM SAF FCDR is to provide such an FCDR of observed brightness temperatures for each individual instrument along with separate inter-calibration offset values to homogenise the observations across all different sensors.

The predecessors of this data record and the data processor suite have originally been developed at the Max-Planck Institute for Meteorology (MPI-M) and the University of Hamburg (UHH) for the Hamburg Ocean Atmosphere Parameters and Fluxes from Satellite Data (HOAPS) climatology. HOAPS is a compilation of climate data records for analysing the water cycle

components over the global oceans derived from satellite observation (Andersson et al., 2011). The main satellite instrument employed to retrieve the geophysical parameters was the SSM/I and much work has been invested to process and carefully homogenise all SSM/I instruments aboard the Defence Meteorological Satellite Program (DMSP) platforms F08, F10, F11, F13, F14 and F15 (Andersson et al., 2010).

The HOAPS processing suite has then been transferred to the Satellite Application Facility on Climate Monitoring (CM SAF)

in a Research to Operations activity in order to provide a sustained environment to process climate data records, which is one of the main tasks of CM SAF (Schulz et al., 2009). The operational processing and reprocessing of the FCDRs and TCDRs, as well as the provision to the research community, is maintained and coordinated by the CM SAF.

The first release of the CM SAF FCDR (Fennig et al., 2013) focused on the SSM/I series, covering the time period from 1987 to end of 2008. In order to continue the HOAPS TCDRs beyond 2008 it was necessary to extend the underlying FCDR

of microwave brightness temperatures with the SSMIS sensor family aboard the DMSP platforms F16, F17, and F18. This was accomplished with the second release of the CM SAF FCDR (Fennig et al., 2015). This combined FCDR of SSM/I and SSMIS brightness temperatures provides a consistent FCDR from 1987 to 2013.

Following requests from users of the FCDR, the third release (Fennig et al., 2017) focused on the extension of the microwave brightness temperature data record to the earlier time period from 1978 to 1987 with observations from the SMMR aboard

Nimbus-7. However, this turned out to be a challenging task, as it has not been possible to get hold of the original raw instrument data records. Instead, the Nimbus-7 SMMR Pathfinder Level 1B Brightness Temperatures data record (Njoku, 2003), available from NSIDC, has been used to generate this FCDR. This work was also a part of data rescue projects aiming to provide FCDRs of historical satellite data to enhance the data coverage for reanalysis projects in the 1960s and 1970s (Poli et al., 2017).





There are also other groups working on the production of FCDRs from SSM/I and SSMIS sensors. Within the Climate Data Record Program of the National Oceanic and Atmospheric Administration (NOAA), two respective FCDRs have been developed. One from the Colorado State University (CSU) (Kummerow et al., 2013) and a second one from Remote Sensing Systems (RSS) (Wentz et al., 2013). Both are using different methods to inter-calibrate the sensors and cover different platforms and time periods. Within the Global Precipitation Measurement (GPM) project, a further data record has been developed with

the aim to utilise all available microwave imagers to provide high quality global precipitation estimates (Berg et al., 2018). This data record is based on the FCDR from CSU (Kummerow et al., 2013) but covers more platforms.

This paper is structured as follows. A short description of the satellite instruments is given in Sect. 2. In Sect. 3 the data processing is explained. This is followed by a description of how the brightness temperatures are derived in Sect. 4 and how the instruments are inter-calibrated in Sect. 5. Uncertainties are discussed in Sect. 6. In Sect. 7 the FCDR is evaluated and

finally the conclusions are provided in Sect. 8. The instrument related specifications are always presented in a logical sequence starting with SSM/I as the most important contributor to the FCDR, followed by the SSMIS and finally the SMMR.

## 2 Instrumentation and data provenance

### 2.1 The SSM/I Instrument

SSM/I sensors have been operated aboard the DMSP satellites as part of the global satellite observing system since 1987. Up

to three satellites have been in orbit simultaneously. An extensive description of the instrument and satellite characteristics has been published by Hollinger et al. (1987) and Wentz (1991). A short summary of the information is given here.

The DMSP satellites operate in a near-circular, sun-synchronous orbit, with an inclination of 98.8° at an approximate altitude of 860 km. Each day, 14.1 orbits with a period of about 102 minutes are performed. The Earth's surface is sampled with a conical scan at a constant antenna boresight angle of 45° over an angular sector of 102.4° resulting in a 1400 km wide swath.

A nearly complete coverage of the Earth by one SSM/I instrument is achieved within two to three days. Due to the orbit inclination and swath width, the regions poleward of 87.5° are not covered.

Six SSM/I instruments have been successfully launched aboard the F08 (June 1987), F10 (December 1990), F11 (November 1991), F13 (March 1995), F14 (April 1997) and F15 (December 1999) spacecrafts. All satellites have a local equator crossing time between 5 and 10 A.M./P.M. for the descending/ascending node. The F08 had a reversed orbit with the ascending node in

the morning. Also, the Earth observing angular sector of the scan on this satellite is, differently from the others, centred to the aft of the sub-satellite track.

The SSM/I is a seven channel total power radiometer measuring emitted microwave radiation at four frequency intervals centred at 19.35, 22.235, 37.0, and 85.5 GHz. All frequencies are sampled at horizontal and vertical polarisation, except for the 22.235 GHz channel, which measures only vertically polarised radiation.

The footprint size varies from 69 km by 43 km with a sampling frequency of 25 km for the 19 GHz channel to 15 km by 13 km with 12.5 km sampling frequency for the 85 GHz channel. The 85 GHz channels are sampled for each rotation of the instrument with a resolution of 128 uniformly spaced pixels, while the remaining channels are sampled every other scan with





a resolution of 64 pixels. A fixed cold space reflector and a reference black body warm load are used for continuous on-board two point calibration.

Main data source for the SSM/I data are the National Environmental Satellite, Data, and Information Service (NESDIS) Temperature Data Records (TDR), available from NOAA's Comprehensive Large Array-data Stewardship System (CLASS). For data earlier than 1993, not covered by the NESDIS TDR, reformatted NetCDF TDR data files from the National Centers for Environmental Information (NCEI) (Semunegus et al., 2010) are used. These are based on SSM/I Antenna Temperature Tapes from RSS (Wentz, 1991) for this time period.

## 100    2.2    The SSMIS Instrument

The SSMIS is the successor to the SSM/I. An extensive description of the instrument and satellite characteristics has been published by Kunkee et al. (2008b). A short summary of essential information is given in this section. The SSMIS instruments are operated, as the SSM/I, aboard the DMSP satellites in an early morning orbit, continuing the existing data record at the same overpass time. The SSMIS has a wider Earth viewing angular sector ($144°$) than the SSM/I, resulting in a 1700 km wide swath.

While four SSMIS have been successfully launched aboard the F16 (October 2003), F17 (November 2006), F18 (October 2009), and F19 (April 2014) spacecrafts, only the first three provide a sufficient data coverage for climate applications. The F19 spacecraft failed about two years after launch.

The SSMIS integrates the imaging capabilities of the SSM/I sensor with the cross-track microwave sounder Special Sensor Microwave Temperature (SSM/T) and Special Sensor Microwave Humidity Sounder (SSM/T-2) into a single conically scan-

ning 24-channel instrument. The SSM/I like frequencies are centred at 19.35, 22.235, 37.0, and 91.35 GHz. All frequencies are sampled at horizontal and vertical polarisation, except for the 22.235 GHz channel, which measures only vertically polarised radiation.

The footprint size varies from 74 km by 47 km with an along-scan sampling frequency of 25 km for the 19 GHz channel to 15 km by 13 km with 12.5 km along-scan sampling frequency for the 91 GHz channels. All channels are sampled for each

rotation, resulting in an along-track sampling of 12.5 km with a resolution of 180 uniformly spaced pixels. The SSM/I-like channels are averaged on board of the spacecraft. Two neighbouring observations are averaged, resulting in a resolution of 90 pixels. A fixed cold space reflector and a reference black body warm load are used for continuous on-board two point calibration.

The data records used as input for the SSMIS are the CSU SSMIS BASE Temperature Data Records (Kummerow et al.,

2013) and the original level 1A TDR data files available from NOAA's CLASS. The BASE data record is a collection of the level 1A unprocessed antenna temperature data that has been written into single orbit granules and reformatted into netCDF-4. These files contain all of the information from the original source TDR files plus additional meta data and updated geolocation information.




## 2.3 The SMMR Instrument

A detailed description of the SMMR is given in Gloersen and Barath (1977) and Madrid et al. (1978). Hence, only a summary of essential information is given here.

Two SMMR instruments were operated, one aboard Nimbus-7 and one aboard SEASAT (Seafaring Satellite). While only about three month of data from the SEASAT mission exist, the SMMR on Nimbus-7 delivered a data record covering nearly eight years from October 25, 1978 until August 20, 1987.

The Nimbus-7 spacecraft operated in a sun-synchronous orbit with an inclination of 99° and an average altitude of 955 km. This configuration results in an orbital period of about 104 minutes and provided approximately 14 orbits per day. The SMMR is mounted on the forward side of the spacecraft in direction of flight. Due to power limitations the SMMR instrument operated most of the time on a 50% duty cycle: one day "on" followed by one day "off".

The SMMR is a ten-channel radiometer, measuring microwave radiation from the Earth's atmosphere and surface in five
frequencies at vertical and horizontal polarisation. Six radiometers were integrated in the instrument, fed by one multi-spectral feedhorn. While the four radiometers at the lower frequencies (from 6.6 GHz to 21 GHz) measured alternating polarisation each half-scan, the other two at 37 GHz measured continuously vertical and horizontal polarisation.

The instruments antenna consists of an offset parabolic reflector and the multi-frequency feed assembly. The reflector is mounted at a nadir angle of 42°, which results in an average Earth incidence angle (EIA) of 50.3°. The antenna rotated within
±25°, centred about the sub-satellite track. This results in a 780 km wide swath at the Earth's surface. The scan velocity varied sinusodially, being fastest at 0° azimuthal scan angle and slowest at the scan edges. A complete scan is accomplished after 4.096 seconds. Cold space is used as the cold reference target, viewed directly through one of three cold-sky calibration horns. The warm calibration target is at the instruments ambient temperature.

Acquisition of Nimbus-7 SMMR data commenced at midnight of October 25, 1978. The SMMR operated continuously
during a three-week checkout period from start-up until November 16, 1978, at which time it began alternate-day operation. This was the normal mode for most of its mission. Under normal operations the SMMR was turned on near midnight GMT (corresponding to a descending equator crossing near 0° longitude) and turned off at approximately the same time the following day, in a continuing sequence. A special operations period occurred from April 3 to June 6, 1986, during which the SMMR was switched on (for ~30 minutes) and off (for ~75 minutes) more frequently. The antenna scan mechanism was turned off
permanently on August 20, 1987 for safety reasons, marking the end of the scanning SMMR data record (Njoku et al., 1998).

The data record used as input for the FCDR is the Nimbus-7 SMMR Pathfinder Level 1B Brightness Temperatures data record, available from the Distributed Active Archive Center (DAAC) at the National Snow and Ice Data Center (NSIDC) (Njoku, 2003).

## 3 Data processing

Generally the data processing is split into a series of processing steps. A generalised data flow diagram is shown in Fig. 1. The required individual processing steps depend on the instrument series and are summarised first in this section to give a general





overview of the processing sequence. The specific steps and corrections are then described in more detail in the following sections.

First, the original input data records are filtered to remove duplicate and erroneous scanlines. The brightness temperatures are reverted to get the original measured counts and a new first guess radiometric calibration is derived (SSM/I, SSMIS). Then the geolocation is performed (SSM/I, SSMIS) and a water-land surface type classification is made. Intermediate level 1a data files in a common data format are produced. These are used to detect sunlight and moonlight intrusion events in the SSMI(S) data records and to produce intrusion correction maps. In the next processing step the individual sensor corrections are applied before the final radiometric calibration is derived. If sensor issues have been identified a specific correction must be applied. These correction methods are either taken from the literature and adapted accordingly or newly developed. The corrections are applied to correct along-scan non-uniformity, SSMI(S) moonlight intrusions, SSMI(S) sunlight intrusions, and SSMIS emission from the main reflector. The corrections are described in detail in Sect. 3.6. After applying the corrections, brightness temperatures and inter-calibration offsets are computed and the results are written to intermediate level 1b data files. This intermediate product is used to compile daily sea ice masks for the SSM/I and SSMIS covered time period. In the last processing step the final surface type classification is performed, the EIA normalisation offsets are computed, an antenna pattern matching is done for the high resolution channels and a final quality control of all Field of Views (FOVs) is applied. The final FCDR data files are compiled in a standardised NetCDF (Network Common Data Form) format, applicable to all sensor types.

## 3.1 Overview

### 3.1.1 SSM/I

A combined usage of different data sources is necessary for the SSM/I, as they cover different time periods and have different coverage due to missing files, checksum errors and corrupt data records. An analysis of SSM/I data from different sources (Ritchie et al., 1998), including NESDIS TDR and RSS data, showed no systematic differences between these data sets. Moreover, our own analysis showed that apart from the geolocation, which is different in the RSS files (Wentz, 1991), the same information is available in all data files.

To merge the scans from all different sources, unique MD5 hash values are computed from the calibration data. The MD5 algorithm is a cryptographic hash function that takes a block of data and returns a unique fixed size hash value that can be used as a fingerprint to identify a specific data block. This MD5 hash value is used in the processing to find identical scan-lines and to correctly merge all available data records to compile the FCDR files. During the first processing step antenna temperatures are reverted to Earth counts using the calibration coefficients provided in the data files. This must be done because the pre-processed TDR input data can contain erroneous calibration slopes and offsets caused by an averaging of the calibration data without quality control tests (Ritchie et al., 1998).

In the following processing step the spacecraft position and the geolocation is computed, the water/land surface type is assigned to each field of view (FOV). A first raw data calibration is done to convert from Earth counts to antenna temperatures





(TAs). From these intermediate level 1a data records yearly monitoring files are compiled where the calibration and geolocation is sampled at fixed orbit angle positions. These files are analysed to identify periods where moonlight (cold view intrusion) or sunlight (hot view intrusion) affect the calibration in order to compile correction tables. They are then used to correct the cold and hot calibration counts. The final calibration coefficients are derived and the antenna temperatures are computed. An along-scan correction is applied to the TAs to account for a non-uniformity of the measured antenna temperatures along a

scan-line. The corrected antenna temperatures are then converted to brightness temperatures and the inter-sensor calibration offsets are computed.

Now the sea-ice concentration is derived from this intermediate level 1b data record for each FOV and daily gridded ice masks are compiled. These maps are then used to assign the sea-ice and sea-ice margin surface types. The Earth incidence angle normalisation offsets are computed for water type FOVs and the high-resolution channels are averaged to provide additional

information matching the antenna pattern at the lower resolution.

### 3.1.2 SSMIS

The SSMIS BASE data record files from CSU have been compiled from the original TDR and were reorganised into single orbit granules with duplicate scans mostly removed. Spacecraft position and velocity have been added by CSU based on North American Aerospace Defense Command (NORAD) orbital data.

Following the procedures developed for the SSM/I, remaining duplicate scan lines in the SSMIS CSU BASE and the original TDR files are first screened and filtered using MD5 hash values of the calibration data. The SSMIS NORAD orbital data sets are not freely available. Therefore the spacecraft position in the CSU BASE data files is used to fit daily orbital data sets, which can then be used in the FOV geolocation procedure. The following processing steps are similar to the SSM/I processing. As an additional step for the SSMIS, an emissive reflector correction is applied for $SSMIS_{F16}$ and $SSMIS_{F17}$.

### 3.1.3 SMMR

The SMMR data record processing can not be done starting from the actual observed Earth counts but has to start from the level 1B Pathfinder data record. Therefore the processing steps differ from the SSMI(S) above.

A detailed description of the level 1B processing steps is given in Njoku et al. (1998) and summarised here for reference. The level 1B Pathfinder data record is a collection of $T_B$ and instrument meta-data, archived as single orbit granules in HDF.

Input to the level 1B processor are level 1A antenna temperatures tapes (TAT). The level 1B data processing was done in daily segments and all algorithms used for running averages were initialised at the beginning of the day and also after data gaps longer than one minute. After reading and decoding, the radiometric calibration was done for each scanline. This was followed by spill-over corrections, an absolute calibration correction, as well as corrections for long-term and short-term drifts. The resulting corrected antenna temperatures were then interpolated along and across scan lines to co-locate all observations to the

37 GHz FOV locations. Finally the polarisation mixing was corrected and a quality status flag was generated for each scan to indicate potential quality losses. This quality flag is also copied to the final FCDR data record for reference.





The CM SAF FCDR processing starts with a further filtering of this level 1B data record. Unique MD5 hash values are computed from the calibration data. All orbit granules for one day are filtered using the MD5 hash values. The data record is also screened for erroneous geolocation and valid scans are merged into daily SMMR FCDR NetCDF files. The scan times are
only available rounded to full seconds. The original time fraction is needed for a correct navigation of the spacecraft position and is therefore estimated with a linear regression fit using all scans from one day. The water/land surface type is assigned to each FOV using the same land-sea-mask as for the SSMI(S) data records.

The radiometer noise equivalent temperatures are estimated from re-computed calibration coefficients for each channel. An along-scan correction is applied to the $T_B$s at 37 GHz to account for a non-uniformity of the measured $T_B$s along the scan-line.
Finally the inter-calibration offsets are computed, and quality flags and global meta-data are assigned.

### 3.2    Geolocation

#### 3.2.1    Geolocation for SSM/I and SSMIS

The quality of the SSM/I geolocation has been examined by Poe and Conway (1990) and Colton and Poe (1999). The operational requirement of geolocating SSM/I data is defined by half the 3dB beam diameter of the high-resolution 85 GHz channels,
which translates to a location error requirement about 7 km. The studies showed that the predicted spacecraft ephemeris in the data records caused an error of up to 15 km. Since July 1989 the operational processing was changed and the new ephemeris error is typically less than 2-3 km. To reduce the SSM/I FOV geolocation error below the requirement a fixed set of spacecraft attitude corrections were determined (Colton and Poe, 1999). These attitude corrections were defined for each instrument. Colton and Poe (1999) finally conclude that the RMS of the geolocation error of the SSM/I FOV is less than 4 km.
The quality of the SSMIS geolocation has been examined by Poe et al. (2008). This study showed geolocation errors in excess of 20-30 km. An automated analysis methodology was developed during the F16 Cal/Val period. As the main sources of the errors, angular misalignments and timing problems have been identified. A set of fixed attitude corrections are found during the Cal/Val period for each SSMIS to reduce the geolocation error to less than 4-5 km.

In order to provide a homogeneous and consistent geolocation in the CM SAF FCDR the attitude corrections must be
applied consistently throughout the entire time period. The geolocation data is therefore re-calculated by CM SAF during the re-processing using the latest available set of corrections. The spacecraft position is predicted with the Simplified General Perturbations model SGP4 (Hoots and Roehrich, 1988). This model uses NORAD ephemeris data sets to predict position and velocity of Earth-orbiting objects. These orbital data is distributed in a two-line element (TLE) data format. The SSM/I element sets for the time period up to January 2000 are freely available and used until December 1999. SSMIS element sets are not
freely available. In order to use the SGP4 model for the entire time period, ephemeris data from the raw data records have been used to find the orbit elements in a minimisation fitting procedure for all SSM/I data files after 2000. For the SSMIS, the spacecraft position, added in the BASE data files, is used to fit the orbit elements.

This fitted ephemeris data is determined for each day and then smoothed using a moving averaging window of $\pm 7$ days. A comparable method is used by Wentz (1991) to fit a simple orbit model. The accuracy is usually better than 1 km compared



to original data and compared to time periods with available NORAD element sets. The spacecraft position is predicted for all scan start times and archived in the FCDR data files.

During the geolocation procedure the spacecraft position is then linearly interpolated to each pixel time in an Earth-centred inertial (ECI) coordinate frame by applying a rotation matrix. The rotation axis is found as the vector cross product between two consecutive spacecraft positions. The magnitude of the position vector is linearly interpolated. From this spacecraft position at
pixel time the location of the FOV on the Earth surface is found as the intersection of the antenna beam with the Earth surface following the geolocation procedure described in Hollinger et al. (1989) and Wentz (1991). While comparing the new and the original geolocation it was found that both agree usually within $\pm 2$ km. The deviations can be attributed to simplifications in the operational processor. The original raw data records (RDR) also show a gradual degradation within a few days followed by sudden jumps when the ephemeris data set was updated in the operational processing.
While the SSM/I had only surface channels, the geolocation procedure for the SSMIS must account for three different groups of feedhorns with different reference heights of the geodetic latitude and longitude. The reference heights are selected similar to the operational algorithm:

- Imaging channels (SSM/I like, channels 12-18) to Earth surface;

- Lower air sounding channels (channels 1-7, 8-11) to 11 km height;

- Upper air sounding channels (channels 19-24) to 60 km height.

New spacecraft attitude corrections for pitch and roll for the SSM/I are found using a method similar to the approach of Berg et al. (2013), where the roll angle is estimated from the slope of the mean $T_B$s along the scan line. Changes in spacecraft roll and pitch have a distinct effect on the vertically polarised $T_B$s: A change in pitch modifies the curvature along the scan and a change in roll results in a gradient along the scan. The method from Berg et al. (2013) has been modified to fit both angle offsets,
roll and pitch, from the $T_B$ variation along the scan. Instead of using only vertically polarised channels for the minimisation procedure as in the original technique (Berg et al., 2013), polarisation differences are employed as the horizontally polarised channels are much less affected. This minimises the influence of cold intrusions at the scan edge. All FOV positions, weighted with the position dependent variability, can be used to find mean values for pitch and roll offsets. After applying the correction, the $T_B$ variation along the scan appears very similar for all instruments. Corrections for yaw and elevation offsets for the
SSM/I instruments are found empirically from a coastline analysis using the high-resolution horizontal 85 GHz channel. The implemented attitude corrections for the SSMIS are taken from the operational configuration (Kummerow et al., 2013) without modifications.

Another geolocation problem that was identified by CM SAF in the raw data records of all SSM/I and SSMIS is an inconsistent handling of leap seconds. During the time period from 1987 to 2015 13 leap seconds are officially inserted to account
for variations in the Earth rotation in order to keep the Coordinated Universal Time close to the mean solar time. It can take up to 7 days before a leap second is introduced to the data record. The time difference itself is not problematic if the observation time and the ephemeris time in the TLE sets are updated synchronously. However, a geolocation error of about 7 km was found





in the original geolocation data if this is not the case. This has been corrected in the CM SAF FCDR and the observation times are changed in a way that all leap seconds are introduced at the correct position.

### 3.2.2 Geolocation for SMMR

The SMMR was observing the Earth surface with a reflector that scanned by oscillating about the vertical axis between azimuth angles of ±25 degrees. Separate radiometers were used for vertical and horizontal polarisation only at 37 GHz. The channels at other frequencies used a single radiometer time-shared between vertical and horizontal polarisation. On the first half of each scan cycle (from right to left), these radiometers measured at vertical polarisation while on the second half of the scan cycle (from left to right) horizontal polarisation was measured. The 37 GHz vertical and horizontal polarisation were measured on both halves of the scan cycle. This original scan pattern and FOV geolocation is not provided in the level 1B data files. Instead all observations were interpolated to the corresponding 37 GHz positions with 47 scan positions for each half-scan. This FOV geolocation remains unchanged in the FCDR data processing.

The actual spacecraft position is used to fit the SGP4 orbit model and to estimate daily TLE sets. The TLE sets from the full data record are completed with TLE sets from http://www.spacetrack.org/, filtered and smoothed using a moving averaging window of ±7 days to compile daily element sets for the entire data record period similar to the SSMI(S) procedures. In the following processing step, the new TLE sets are now used to predict the spacecraft position of each scan line. If the predicted and the archived position are offset by more than 6 km, the scan is marked in the scan quality flag with a geolocation error. Fractional revolution number and local azimuth angles (not available in the level 1B files) are computed using the predicted spacecraft position. Also scan angles, reflected sun-footprint angles, and EIA are only available at 30 fixed positions per scanline in the level 1B data record. These variables are interpolated to each of the 94 FOV full scan positions.

The SMMR data record contains erroneous data, which has to be flagged in a first processing step during the geolocation. The spacecrafts attitude information (roll, pitch, yaw angles) is scanned for zero filled values and outliers. These angles were used without quality control in the level 1B processing to compute the EIAs. Errors in the EIA are mostly caused by anomalies in the attitude angles. Therefore the zero filled values and outliers in the attitudes angles and the EIA are replaced with interpolated values. However, no extrapolation is applied and incorrect values at the beginning or end of the data files are set to undefined. The original quality flag status is unset accordingly in this case as the original flagging is no longer necessary due to the interpolation.

### 3.3 Antenna pattern matching

For the application of geophysical retrieval algorithms it is important that about the same area of the Earth surface is seen by all channels. Due to different, frequency dependent, FOV sizes and sampling patterns, the covered area can be significantly different. Also the FOV positions can be shifted along and across track because the SSMIS feedhorns are not exactly co-aligned. For convenience in the application of geophysical retrievals, the high-resolution channels at 85 GHz (SSM/I) and at 91 GHz (SSMIS) are averaged down to the resolution and position of the corresponding 37 GHz FOV. These channels are made available in both, the original high resolution and sampling as well as in the lower resolution and sampling of the 37 GHz





channel in the FCDR data files. FOV differences between the lower resolution channels at 19, 22, and 37 GHz are not accounted for.

### 3.3.1 SSM/I

An SSM/I FOV at 85 GHz covers about 20% of the area sampled at 37 GHz. The sampling distance for the 85 GHz channel is
12.5 km and for the 37 GHz channel 25 km. The high frequency channels are thus sampled twice as often in along and across track direction compared to all other SSM/I channels. In order to get a comparable coverage with the 85 GHz channels, nine neighbouring pixels of the high resolution scan lines are averaged to match the size of the corresponding 37 GHz pixel. The high resolution 85 GHz $T_B$s are averaged with a Gaussian weighted distance $w$ from the centre FOV resembling the 37 GHz main antenna pattern:

$$
\begin{aligned}
\quad T_{B,lo}(s,c) = \sum_{i=-1}^{1} \sum_{j=-1}^{1} w(c,i,j) \cdot T_B(hs+i, hc+j) \\
\Big/ \sum_{i=-1}^{1} \sum_{j=-1}^{1} w(c,i,j)
\end{aligned} \tag{1}
$$

where $hs = s \cdot 2$ and $hc = c \cdot 2$ are the high resolution along-track and across-track indices respectively and $s$ and $c$ are the corresponding low resolution indices. The Gaussian weights for the surrounding 8 pixels depend on the relative position of the centre pixel along the scan line $c$ due to the distortion of the conical scanning system at the scan edges. An array of pre-
calculated averaging weights is used in the data processing, providing a fixed set of 9 specific weighting coefficients for each of the 64 FOVs in the lower SSM/I resolution.

### 3.3.2 SSMIS

Due to the higher frequency, a single FOV at 91 GHz covers only about 17% of the area sampled at 37 GHz. The lower resolution channels are averaged on board of the spacecraft, combining two neighbouring observations. Hence, the 91 GHz
channels are sampled twice as often in along scan direction compared to the 37 GHz channels and the actual centre positions are no longer co-registered with the non-averaged high resolution FOV positions. In order to get a comparable coverage with the 91 GHz channels, 6 neighbouring beam positions (2 along scan, 3 along track) are averaged to match the resolution of the corresponding 37 GHz footprint. The averaging is done similar to Eq. (1), by weighting the individual brightness temperatures with their distance $w$ from the centre FOV.

### 3.4 Land mask and sea ice detection

Each FOV is characterised with a surface type classification flag using CF-Metadata convention for flag variables. Possible surface types are: water, land, coast, sea-ice, and sea-ice margin. The centre latitude and longitude of the FOV is used to assign the surface type.



The FCDR specific land-sea mask is derived from the Global Land One-km Base Elevation data base (GLOBE Task Team,
1999). This data set is further adjusted to the footprint resolutions by first removing small islands and landmasses with a
diameter of less than 5 km for the low-resolution channels and 2 km for high-resolution channels, treating these areas as open
water. This provides the basic land-sea classification. In a second step the coastal areas are defined by expanding the remaining
land areas 50 km (low resolution) or 15 km (high resolution) into the sea.

To account for the varying sea ice margins, a daily sea-ice mask is compiled from the observed $T_B$s. These maps are created
in two steps. First the total sea ice covered fraction within a single FOV is computed using the NASA Team sea ice algorithm
of Swift et al. (1985). The resulting sea ice observations from all available instruments are then gridded to common daily mean
fields on a regular 0.5° x 0.5° grid. In order to distinguish between short-lived strong rain events and persisting sea ice, which
are characterised by similar temperature signatures, only grid boxes with an average sea ice fraction above 15% for at least 5
consecutive days are flagged as ice covered. This filter removes all events lasting less than 5 days from the first guess ice mask.
Daily sea ice maps are then derived from this screened data set by re-expanding the reliably identified sea ice areas in time and
space while filling remaining data gaps by spatial and temporal interpolation. Finally, the resulting sea ice margin is extended
50 km into the ocean which leads to the sea-ice and sea-ice margin surface types.

## 3.5  Radiometric calibration

The basic assumption for a microwave radiometer calibration is a linear relation of the radiometer output voltage (measured in
counts) to the radiometric input, neglecting non-linear effects in the detector. Two reference targets with known temperatures
and corresponding radiometer measurements at these temperatures are sufficient for an absolute linear two-point calibration.
The actual implementation of the radiometer differs between the SMMR and SSMI(S) instruments due to the different design.

### 3.5.1  SSM/I

The SSM/I calibration equation is defined as follows (Hollinger et al., 1987):

$$T_A = S \cdot C^e + O \,, \tag{2}$$

where $T_A$ is the antenna temperature, $C^e$ are the measured counts when the radiometer is switched to the antenna (Earth view),
$S$ is the calibration slope and $O$ is the offset. The warm calibration target of the SSM/I is an internal black-body radiator with
three embedded thermistors, constantly measuring its temperature. The cold reference target is a mirror pointing to the cold
space, with an assumed constant radiometric background temperature at 2.7 K. The radiometer views the internal warm and
cold calibration targets once during a scan rotation. For all seven instrument channels five radiometric readings are taken for
both calibration targets during an A-scan and five additional measurements are taken for the two 85 GHz channels during a
B-scan. The three embedded thermistor readings are available once per scan pair. The calibration slopes $S$ in K/count and





offsets $O$ in K for each instrument channel are calculated following Hollinger et al. (1987) as follows:

$$S = \frac{\langle \overline{T^h} \rangle - T^c}{\langle \overline{C^h} \rangle - \langle \overline{C^c} \rangle} \,,$$


$$O = \frac{T^c \cdot \langle \overline{C^h} \rangle - \langle \overline{T^h} \rangle \cdot \langle \overline{C^c} \rangle}{\langle \overline{C^h} \rangle - \langle \overline{C^c} \rangle} \,, \tag{3}$$

where:

$\langle \overline{C^h} \rangle$ = Smoothed mean warm counts,

$\langle \overline{C^c} \rangle$ = Smoothed mean cold counts,

$\langle \overline{T^h} \rangle$ = Smoothed mean warm target temperature,

$\quad T^c$ = Cold target temperature.

The smoothed scan-line mean values for the cold counts $C^c$, the warm counts $C^h$, and the warm target temperature $T^h$ are derived by first averaging all available measurements for each scan pair that pass the quality control:

$$\overline{C^h} = \frac{1}{n} \cdot \sum_{s=1}^{n} C^h_s \,,$$

$$\overline{C^c} = \frac{1}{n} \cdot \sum_{s=1}^{n} C^c_s \,, \tag{4}$$


$$\overline{T^h} = \frac{1}{3} \cdot \sum_{s=1}^{3} T^h_s \,,$$

where $C^c_s, C^h_s, T^h_s$ are the individual samples of the warm target, the cold target, and the warm load target thermistors respectively. In order to further reduce the noise in the calibration input values, an additional Gaussian smoothing of these scan-line mean values, denoted with the $\langle \, \rangle$ operator is applied:

$$\langle \overline{C^h} \rangle = \sum_{i=s-g}^{s+g} w_i \cdot \overline{C^c}_i \bigg/ \sum_{i=-g}^{g} w_i \,,$$


$$\langle \overline{C^c} \rangle = \sum_{i=s-g}^{s+g} w_i \cdot \overline{C^c}_i \bigg/ \sum_{i=-g}^{g} w_i \,, \tag{5}$$

$$\langle \overline{T^h} \rangle = \sum_{i=s-g}^{s+g} w_i \cdot \overline{T^h}_i \bigg/ \sum_{i=-g}^{g} w_i \,,$$

where $w_i$ are the Gaussian across-scan weighting coefficients. The moving averaging kernel spans the $g$ preceding and $g$ following mean scan-line values. The kernel half size $g$ is set to 5 for all SSM/I sensors. This smoothing process introduces an artificial error correlation in the smoothed calibration coefficients. However, as the variance of the calculated $T_B$s is dominated

by the variance in the independently measured Earths counts $C^e$, the impact of the error correlation in the smoothed calibration coefficients on the final brightness temperatures can be neglected.





The temperature of the warm target $T^{hl}$ needs to be corrected using the temperature of the forward radiator plate $T^{pl}$ to account for radiative coupling between the warm load and the top plate of the rotating drum assembly which faces the warm load when not being viewed:

$$T^h = \varepsilon \cdot T^{hl} + (1 - \varepsilon) \cdot T^{pl} \, . \tag{6}$$

The value for $\varepsilon$ is empirically determined from thermal-vacuum calibration (Hollinger et al., 1987). The original value of $\varepsilon = 0.99$ has been changed to sensor specific values determined during the inter-sensor calibration procedure (cf. Sect. 5) for this FCDR. Calculated slopes and offsets are archived in the FCDR data files for traceability. The antenna temperatures are then calculated from the Earth counts $C^e$ for each FOV using Eq. (2). Keeping the calibration coefficients in the data record

allows to revert the original Earth counts from the temperatures.

### 3.5.2 SSMIS

The calibration of the SSMIS follows the SSM/I calibration with Eq. (2) and Eq. (3). The cold and warm calibration view counts and the warm target temperature are already averaged by the SSMIS on-board software and are down-linked as scan-line mean values. The across-scan averaging kernel half width $g$ in Eq. (5) is set to eight scan-lines for channels 1-7, four

scan-lines for channels 8-18, and 32 scan-lines for channels 19-24. This setup also corresponds to the latest operational ground processing software configuration (Kummerow et al., 2013).

The configuration of the on-board averaging software has changed over time for SSMIS$_{F16}$ and SSMIS$_{F17}$. The original setup was to average the radiometric readings of the cold and warm target of the current scan and the seven preceding scan-lines, and thus asymmetric. Because this configuration resulted in a striping pattern, the on-board across-scan averaging has

been eventually switched off after revolution number 29808 for SSMIS$_{F16}$ and revolution number 1062 for SSMIS$_{F17}$. In order to get a symmetric average of the calibration counts for the time period when only the asymmetric mean values are available, the averaging window for channels 8-18 must be expanded to a minimum of 14 scans. This results from averaging the two 8-scan mean values at scan $n$ and at scan $n + 7$. This larger averaging kernel size reduces the noise in the affected SSMIS$_{F16}$ and SSMIS$_{F17}$ channels during the aforementioned time period.

### 3.5.3 SMMR

The calibration method applied in the Pathfinder level 1B data record for the SMMR was taken from the SMMR CELL data record (Fu et al., 1988):

$$T_A = I + S \cdot \left( \frac{C^e - C^h}{C^c - C^h} \right) \, , \tag{7}$$

where $T_A$ is the antenna temperature, $C^e$, $C^h$, and $C^c$ are the measured counts when the radiometer is switched to the antenna

(Earth view), the warm load calibration, and cold load calibration, respectively. The calibration coefficients $S$ and $I$ are defined



as functions of the instruments component temperatures:

$$I = a_1 + a_2 T_{sw} - \Delta_{fh}$$
$$S = a_3 \left( T_{sw} - 2.7 \right) + \Delta_{ch} . \tag{8}$$

The $\Delta_{fh}$ term is a correction for temperature gradients from the multi-frequency feedhorn along the feedhorn waveguide to the receiver and is given by:

$$\Delta_{fh} = \left( T_{sw} - T_{fh} \right) - \left( \frac{T_{fw} - T_{fh}}{\alpha_1} \right) + \left( \frac{T_{sw} - T_{fw}}{\alpha_1 \beta_1} \right) , \tag{9}$$

with $T_{sw}$, $T_{fh}$, and $T_{fw}$ are the Dicke switch, feedhorn, and feedhorn waveguide temperatures, respectively.

The $\Delta_{ch}$ term is a correction for temperature gradients from the cold-sky calibration horn along the calibration horn waveguide to the receiver and is given by:

$$\Delta_{ch} = \left( T_{sw} - T_{ch} \right) - \left( \frac{T_{cw} - T_{ch}}{\alpha_2} \right) + \left( \frac{T_{sw} - T_{cw}}{\alpha_2 \beta_2} \right) , \tag{10}$$

with $T_{ch}$ and $T_{cw}$ are the calibration horn and calibration horn waveguide temperatures. The calibration coefficients $a_{1,2,3}$, $\alpha_{1,2}$, and $\beta_{1,2}$ are given in Njoku et al. (1998).

The warm calibration target of the SMMR is the internal ambient temperature at $\approx$300 K. The cold reference target is the cold space, viewed with three calibration horns (depending on frequency). The cold sky is assumed to be at a constant temperature of 2.7 K, i.e. cosmic background temperature. The radiometer views the internal warm and cold calibration targets once during a full scan cycle. The instrument temperatures are sampled every eight scans using platinum resistance thermistors and are embedded in the data record.

In order to reduce the noise, calibration warm and cold counts are smoothed in the level 1B data processing using a running one-sided average:

$$\left\langle C^{h,c}(s) \right\rangle = w \cdot C^{h,c}(s) + (1 - w) \cdot \left\langle C^{h,c}(s-1) \right\rangle , \tag{11}$$

with $s$ as the scanline number, $w$ as the weight of the new data value and $\left\langle C^{h,c} \right\rangle$ as the across-scan smoothed cold or warm count. A weight of $w = 0.1$ was chosen as a compromise between noise reduction and overdamping.

Also the instrument temperatures are averaged using the same method. As these temperatures are only available every eight scans, a weight of $w = 0.6$ was used.

The calibration coefficients $S$ and $I$ are not available in the Pathfinder level 1B data records. In order to complete the data record and to estimate the instrument noise level, these coefficients are re-calculated using Eq. (8) to Eq. (10) and then archived in the SMMR FCDR data files for reference.

## 3.6 Corrections applied to raw data records

A number of technical issues affecting the quality of the measurements have been identified for all radiometers. They have to be corrected as one major part of the FCDR generation before an inter-calibration can be derived.



### 3.6.1 SSM/I along-scan correction

One of the first detected SSM/I calibration problems was a non-uniformity of the measured antenna temperatures along a scan-line (Wentz, 1991; Colton and Poe, 1999). A rapid fall-off at the end of the Earth scan of nearly 1.5 to 2 K was found in all SSM/I channels. This systematic problem is caused by an intrusion of cosmic background energy by the glare Suppression System-B (Colton and Poe, 1999) into the antenna feedhorn reducing the observed scene temperature and thus causing a scan-position and scene temperature depending offset.

This intrusion effect can be treated as an antenna beam position energy loss. It can be corrected by multiplying the antenna temperature $T_A$ with a scan position dependent correction factor. This factor is derived in a two step procedure. First, antenna temperatures for each sensor and channel with surface type classified as sea between 50°S and 50°N, thereby excluding areas with seasonally varying sea-ice, are normalised to a constant Earth incidence angle (EIA). Then these temperatures are averaged into FOV-position bins for at least one year. The correction factor at each FOV position is computed as the ratio of the position averaged value to the average along the unaffected FOVs positions at the scan line centre (FOV positions 50-90). The derived along-scan correction factors are very similar for all SSM/I sensors. As an example, the along-scan correction for SSM/I$_{F13}$ is shown in Fig. 2.

### 3.6.2 SSM/I cold space reflector intrusions

The SSM/I calibration procedure assumes that the cold reference target is at the background cosmic temperature of 2.7 K. However, due to the open instrument design it is possible, depending on the orbit parameters, that the moon is visible in the cold view. This leads to a short term increase in the measured cold target radiation counts. The amplitude and length of this increase depends on the beamwidth of the channel as the fraction of the moon in the FOV compared to the cosmic background is larger for smaller beamwidths. These events happen usually twice per month, depending on the orbit configuration. An example is shown in Fig. 3. The image depicts two intrusion events, the first one over the North Pole at the 14th and the second one at the 19th November shortly before crossing the South Pole.

As these events only affect the cold reference target, the impact on the calibrated antenna temperatures is scene dependent and larger for lower scene temperatures. The best way to correct this effect is to apply a band-pass filter to remove the additional signal induced from the moonlight. Under normal conditions the variation of the cold counts is periodic and changes very little from orbit to orbit. This allows to easily detect the anomalies as shown in Fig. 3.

A procedure has been developed which automatically detects the moonlight intrusion events using a Laplace filter as an edge detection algorithm. The Laplace operator computes the second derivatives of an image, measuring the rate at which the first derivatives change. This determines if a change in adjacent pixel values is from an edge or continuous progression. In order to use standard image filtering techniques to detect the intrusion events, yearly monitoring files are compiled for each sensor and channel. The calibration data is sampled at fixed orbit angle positions. The discrete cold count Laplace operator $\Delta^c$ is then defined as:

$$\Delta_{i,j}^c = 2 \cdot \hat{C}_{i,j}^c - \hat{C}_{i-k/2}^c - \hat{C}_{i+k/2}^c \,, \tag{12}$$





with $i$ and $j$ are the along-orbit and across-orbit image indices respectively, $k$ is the kernel size, and $\hat{C}^c$ are the sampled cold counts. This image is then smoothed and periods with moonlight intrusion are detected where the derived Laplacian is significantly larger then one standard deviation of the background field. These periods are regarded as missing data and reconstructed along the orbit using a Spline interpolation procedure from the neighbouring smoothed valid values (see Fig. 3 cyan line). The difference between the original and the reconstructed values is smoothed and removed from the original values. This procedure works similar to a local band-pass filter but can also deal with undefined data. Corrected scan lines are also flagged during the final data processing using a processing control flag.

### 3.6.3  SSM/I warm load intrusions

Another problem in the SSM/I calibration data that has been identified is sunlight intrusion into the warm reference black body target. This occurs during certain orbit constellations when the satellite platform is leaving and entering the Earth shadow.

The issue is caused by an illumination of the black body tines due to either direct or reflected sunlight. The sunlight causes a fast instant response in the calibration measurements of the SSM/I as the radiation heats the tines of the black body while the black body core remains unaffected or is just slowly heating up. As the warm load temperature sensors are embedded into the core of the black-body, they do not measure the temperature of the calibration target at the tines but the core temperature. With the delayed response of the temperature sensors to the increased radiation, the net effect is a sharp artificial discontinuity (decrease) in the calibration gain. Figure 4 shows an example for the detection of intrusion events. The intrusions occur around the orbit positions 0.23 and 0.81. When the satellite leaves the Earths shadow, the whole instruments warms up and also the warm load temperature increases. When the intrusion ceases, the black body core and surface temperature are balanced again (around position 0.32). However, if the satellite enters the Earths shadow the intrusion effect is more pronounced. As long as the sunlight falls on the warm load, a negative anomaly can be observed in the calibration slopes, because the core temperature of the black-body remains nearly constant. When the sunlight intrusion vanishes (around position 0.9) a sharp increase is visible in the slope when measured warm load target temperature and the observed radiation are balanced again.

This sunlight intrusion effect depends on the channel frequency and is strongest at the lower frequencies. The procedure to correct for this intrusion is similar to the correction for the moonlight intrusions, but in the sunlight case the warm target counts are affected. All significant sunlight intrusion events are detected using the edge-detection algorithm applied to the slope anomalies. The discrete Laplace operator $\Delta^s$ is applied to the sampled calibration slopes $\hat{S}$:

$$\Delta^s_{i,j} = 2 \cdot \hat{S}_{i,j} - \hat{S}_{i-k/2} - \hat{S}_{i+k/2} , \tag{13}$$

The intrusion periods are filtered and reconstructed from a spline interpolation using the neighbouring smoothed valid slope values (see Fig. 4 cyan line). The difference between the original and the reconstructed slope values is smoothed to obtain the intrusion induced anomaly and then used to correct the computed slopes. Finally the corrected slopes and the warm load calibration temperature are used to convert the slope anomaly into warm load count offset maps. These corrections are applied during the final data processing to correct the intrusion events. Affected scans are also flagged using a processing control flag.

An example for the edge detection algorithm (applied Laplace operator) is shown in Fig. 4. The positions of the sunlight intrusion events are identified as positive and negative anomalies lines. The geographic positions of the intrusions depict a seasonal cycle caused by the varying relative angle of the satellite to the sun. The pattern also changes from year to year due to the orbit drift and changing equator crossing times. Therefore the correction maps are compiled for each year separately.

### 3.6.4 SSMIS along-scan correction

Similar to the non-uniformity of the measured antenna temperatures along a scan line for the SSM/I, also the SSMIS sensors show a rapid decrease near the end and begin of the Earth viewing scene $T_B$s (Kunkee et al., 2008b). A rapid falloff at the end of the Earth scan is found for channels 15 and 16 (37 GHz), due to intrusions into the feedhorn by the warm load cover and for the beginning of the scan line for channels 12-14 (19 and 22 GHz), due to intrusions by the cold-sky reflector. The strongest effect can be observed for the SSMIS$_{F17}$ channels 15-16, when the last 5 beam positions are biased by more than 8 K.

This effect is corrected following the approach for the SSM/I by multiplying the antenna temperature with a scan position dependent correction factor. The along-scan correction for the SSMIS$_{F16}$ instrument are exemplarily shown in Fig. 5.

### 3.6.5 SSMIS cold space reflector intrusions

Due to the heritage design from the SSM/I, also the moonlight intrusions into the cold space reflector are inherited by the SSMIS. The correction procedure principally follows the one from the SSM/I. However, the SSMIS consists of six individual feedhorns with slightly different alignments. This leads to a small shift in the location of the intrusion, which has to be accounted for and the correction is carried out for each feedhorn separately.

### 3.6.6 SSMIS warm load intrusions

As with the cold space reflector intrusions, the sunlight intrusion into the warm calibration black body target are also inherited by the SSMIS. In fact, the warm load intrusion were detected first in the SSMIS$_{F16}$ (Kunkee et al., 2008a) because they are stronger and cover a significantly larger part of the orbit. Kunkee et al. (2008a) identified two intrusion regions where the sun directly shines on the warm load tines and two regions where the radiation is reflected from the top of the SSMIS canister onto the warm load target. In order to mitigate the direct intrusions, a fence was installed aboard F17 and F18.

Following the approach developed for the SSM/I, the sunlight intrusion events for the SSMIS are detected by applying the Laplace operator Eq. (13) to the calibration slope. An example showing 4 sunlight intrusion events at orbit positions 0.04, 0.15, 0.48, and 0.65 is presented in Fig. 6 for SSMIS$_{F16}$. Apart from the regularly repeated short-term moonlight intrusion events, which are occurring around orbit positions 0.3 and 0.7, strong regular stripes can be identified in Fig. 6a. The negative anomaly regions in the Laplacian, flanked with positive anomalies, are the regions with sunlight intrusions affecting the calibration gain. These patterns are not static at a geographic position, but vary during the year as a function of the solar elevation and azimuth angles relative to the spacecraft platform. Also spacecraft structure elements, like solar panels, can shade the warm load during some periods of the year. This can be observed in February (March) around orbit position 0.5, where the anomaly





abruptly stops (starts). The additional fence on DMSP F17 and F18 is working well. Only during a three month period, when the sun is above the installed fence, an intrusion is visible for SSMIS$_{F17}$ and just a small intrusion region for SSMIS$_{F18}$ can be identified throughout the year (not shown here). The basic correction method again follows that of the SSM/I. However, as

a consequence of the aforementioned factors, it is not possible to compile one static intrusion map for each channel, instead yearly correction maps are compiled for each channel. In order to minimise the noise, the SSM/I procedure has been adapted to convert the observed count anomalies into an anomaly in the warm target temperature first. These temperature anomalies are then averaged across all channels sharing one feedhorn and eventually converted back to channel specific count anomalies.

### 3.6.7  SSMIS emission from main reflector

The SSMIS main reflector appears to be emissive (Bell, 2008; Kunkee et al., 2008a), which results in an additional radiation emitting from the reflector at the reflectors temperature and thus modifying the measured scene temperature. The magnitude of this effect depends on the temperature difference between the reflector and the Earth scene. The reflector temperature varies along the orbit due to solar heating when not being covered in the Earth shadow. The strongest signal can be observed when the spacecraft leaves the Earth shadow and the reflector temperature increases by about 80 K within a few minutes.

If the temperature of the main reflector $T_{refl}$ and its emissivity $\varepsilon_{refl}$ are known, the emission effect can be corrected with:

$$T'_A = \frac{T_A - \varepsilon_{refl} \cdot T_{refl}}{1 - \varepsilon_{refl}} \; . \tag{14}$$

Values for $\varepsilon_{refl}$ were found by comparing observed minus background residuals from data assimilation experiments by Bell (2008). Emissivity values derived for the Universal Preprocessor (UPP) version 2 are applied in this FCDR unchanged for all non SSM/I like channels. Emissivity values for the SSM/I like channels are fitted again during the inter-calibration minimisation

procedure because the given values turned out to be not at the required precision. The values are determined by using the SSMIS$_{F18}$ as the reference target, because no significant emissivity was found for its reflector.

From SSMIS$_{F17}$ onward, a temperature sensor is placed at the back of the reflector to measure $T_{refl}$ directly, but for SSMIS$_{F16}$ the closest available sensor is placed on the rim of the main reflector, close to the support arm. The temperature of the reflector can be constructed from the arm temperature $T_{arm}$ using a lagged correction following (Bell, 2008). The

reconstructed reflector temperature is computed for F16 and saved in the FCDR data files. The reflector temperature for all sensors is linearly interpolated to the observation time as the housekeeping data records are not available for each scan line and the antenna temperatures are corrected with Eq. (14).

### 3.6.8  SMMR Pathfinder level 1B corrections

In the level 1B processing, corrections for absolute calibration, long-term drifts, short-term drifts, and polarisation mixing

are applied. The level 1B corrections applied to the antenna temperatures are explained in detail in Njoku et al. (1998) and summarised in the following two subsections.





**(a) SMMR calibration offset adjustments**

A two-point linear absolute calibration adjustment to the SMMR data record was estimated by Njoku et al. (1998) using ocean areas acting as the cold calibration target and the internal ambient instrument warm load as the warm calibration target. The coefficients were derived over calm, cloud-free oceanic areas, characterised by low emissivity values. Modelled $T_B$s were compared to the observed SMMR $T_B$s to derive a linear calibration adjustment to the antenna temperatures:

$$T'_{Ac} = a + b \cdot T'_A \,, \tag{15}$$

where $T'_A$ is the normalised SMMR antenna temperature (after spill-over correction), $T'_{Ac}$ is the corrected antenna temperature value, and $a$, $b$ are the linear calibration adjustment coefficients. The calibration constraints can be written as:

$$T^m_B = T^o_{Bc} = a + b \cdot T^o_B \,, \tag{16a}$$
$$I' = a + b \cdot I', \tag{16}$$

where $T^o_B$ is the mean observed SMMR brightness temperature for the cold calibration target and $T^m_B$ is the corresponding modelled brightness temperature. Solving these equations for the coefficients $a$ and $b$ yields:

$$a = I' \cdot \frac{T^m_B - T^o_B}{I' - T^o_B} \,,$$
$$b = \frac{I' - T^m_B}{I' - T^o_B}. \tag{17}$$

The calibration correction coefficients $a$ and $b$ depend on the warm load equivalent brightness $I'$ and are updated online for each new set of calibration data. The values for $T^m_B$ and $T^o_B$ are given in Njoku et al. (1998).

**(b) SMMR correction for short and long-term drifts**

Ageing of the radiometer components can lead to a slow but steady degradation in the radiometer performance. Njoku et al. (1998) studied the long-term evolution of the mean $T_B$s in all SMMR channels. The largest drift was found for the channel at 21 GHz,v-pol with an increase of 36 K until March 1985 when it was turned off. The observed drifts are corrected in the level 1B data record using a polynomial fit of the globally averaged antenna temperatures, after removing mean seasonal cycle variations. The polarisation mixing was not corrected for in order to keep the signals from each radiometer independent at this stage. Otherwise the strong drift in the horizontal polarisation at 21 GHz would have been mixed into the vertical polarised channel.

Also orbit dependent solar illumination of the instrument can affect the radiometric calibration. Although temperature gradients are accounted for in the SMMR calibration Eq. (7) and Eq. (8), not all effects might be compensated accurately. Especially when the spacecraft leaves or enters the Earth's shadow, strong temperature gradients can be observed, possibly leading to un-modelled thermal emissions in the wave guides and switches of the instrument (Francis, 1987).

To correct for this type of errors, orbit dependent corrections were derived following a method described by Francis (1987). Mean brightness temperature differences between ascending and descending orbits were analysed to derive $T_B$ drift corrections





as a function of the spacecraft ecliptic angle. The corrections are derived for each year and then linearly interpolated to the observation time.

The corrections for long-term drifts $T_{lt}$ and short-term drifts $T_{st}$ were incorporated into the calibration by replacing the calibration correction coefficients $a$ and $b$ in Eq. (17) with drift corrected coefficients $a_n$ and $b_n$. However, as the original coefficients $a_n$ and $b_n$ are not available in the data records and the short-term and long-term correction maps are not available in Njoku et al. (1998), it is not possible to undo the applied corrections and to revert the original antenna temperatures. Therefore, any other additional correction must be done on top of the already applied level 1B corrections.

### 3.6.9 SMMR along-scan correction

A general along-scan non-uniformity of the SMMR antenna temperatures is to be expected, due to the design of the radiometer, causing a scan angle dependent cross-polarisation coupling. Although most of this effect is removed by the applied scan-position dependent polarisation mixing correction (see Sect. A3), significant variations in the final $T_B$s can remain. To analyse these, the match-up dataset with observed and modelled brightness temperatures from the inter-calibration procedure (see Sect. 5.3) has been used here. Differences between observed and modelled $T_B$s for each channel with surface type classified as sea between $50°S$ and $50°N$ are averaged into FOV-position bins for the complete SMMR time period at 10-daily intervals. Then a factor at each FOV position is computed as the ratio of the position averaged value to the unaffected FOVs at the scan line centre (azimuth$= 0$). The time mean results are shown in Fig. 7 for a nominal scene temperature at 200 K. Most pronounced features are the large offsets of about 1 K for the channel at 21 GHz,v-pol on the left scan side and for the channel at 37 GHz,v-pol on the right scan side. The other channels show small offsets on the left side (0-0.3 K) and small but a little larger offsets at the right side (0.2-0.5 K). The large differences at 21 GHz,v-pol are not constant over time (not shown here) and are most likely caused by the polarisation mixing from the problematic channel at 21 GHz,h-pol. This channel depicts a strong trend and degradation with time, which can not be removed completely by the long-term drift corrections and polarisation mixing decoupling procedure. Also the observed differences in the 18 GHz channels are not constant over time, but do show a seasonal variation with smaller offsets in northern hemisphere winter and larger offsets in northern hemisphere summer months. Only the deviations in the 37 GHz channels are constant over time.

Following these results, only the along-scan biases in the 37 GHz channels are corrected in the CM SAF SMMR FCDR. As the differences in the vertical polarisation are much larger than in the horizontal polarisation, the cross- polarisation is assumed to be zero and the bias correction reduces to a scan position depended factor similar to the SSMI(S) correction procedure.

## 4  Computation of brightness temperatures

When a space-borne radiometer is observing the Earth surface, the antenna boresight direction can be defined as the unit vector $\hat{s}_0$ from the spacecraft to the Earth surface. The brightness temperature $T_B$ at a polarisation $p$ propagating to the antenna from a surface point $\rho$ is then defined as $T_{B_p}(\rho, \hat{s})$, where $\hat{s}$ is the unit vector from the antenna to surface at an angle $d\Omega$. Following Stogryn (1978), the corresponding antenna temperature $T_{A_p}(\hat{s}_0)$ can be expressed as an integral of the normalised co-polarised





antenna pattern $G_{pp}(\hat{s_0}, \hat{s})$, and cross-polarised antenna pattern $G_{pq}(\hat{s_0}, \hat{s})$:

$$T_{A_p}(\hat{s_0}) = \int\limits_{4\pi} \Big[ G_{pp}(\hat{s_0}, \hat{s}) T_{B_p}(\rho, \hat{s}) +$$
$$G_{pq}(\hat{s_0}, \hat{s}) T_{B_q}(\rho, \hat{s}) \Big] d\Omega \qquad (18)$$

where $q$ is the polarisation orthogonal to $p$.

The measured antenna temperature $T_A$ is therefore directly related to the radiation entering the antenna feedhorn as observed with the effective co- and cross-polarised far-field antenna power patterns. To obtain a correct estimate of the actual scene

brightness temperature $T_B$ from the measured $T_A$ it is necessary to correct for the radiation from the cross polarisation (cross-polarisation leakage $\chi_{v,h}$) and from the cold-space (feedhorn spillover loss $\Lambda_{sp}$) (Hollinger et al., 1987). These corrections are generally referred to as antenna pattern correction (APC). In order to minimise systematic uncertainties in the computed brightness temperature data record, the antenna specifications must be carefully characterised before launch.

Introducing $\phi$ as the angle between the antenna polarisation vector and the Earth polarisation vector with the antenna gain

as $\sin^2 \phi$ dependence (Njoku et al., 1998) Eq. (18) can be written as:

$$T_{A_p} = \int\limits_{4\pi} \Big[ G_{pp}(T_{B_p} \cos^2 \phi + T_{B_q} \sin^2 \phi) +$$
$$G_{pq}(T_{B_q} \cos^2 \phi + T_{B_p} \sin^2 \phi) \Big] d\Omega \ . \qquad (19)$$

This integral can be sub-divided into an Earth-view region $\Omega_e$ and a cold-space region $\Omega_{sp}$. It is assumed that the cold-space cosmic background $T^c$ is constant at 2.7 K. The cold-space contribution

$$\Lambda_{sp} = \int\limits_{\Omega_{sp}} (G_{pp} + G_{pq}) d\Omega \qquad (20)$$

is subtracted from the antenna temperature while the antenna gains are re-normalised over the Earth-view region to derive the Earth antenna temperature $T'_{A_p}$:

$$T'_{A_p} = \frac{T_{A_p} - T^c \cdot \Lambda_{sp}}{1 - \Lambda_{sp}} \ . \qquad (21)$$

This is also referred to as spillover correction or space side-lobe correction and is applied for all instruments. The spillover

fractions for the SMMR are taken from Fu et al. (1988), for the SSM/I from Wentz (1991), and for the SSMIS from the corresponding ground processing software.

A detailed description of the derivation of the equations to compute the $T_B$s for each individual sensor is given in Appendix A.





## 5   Sensor inter-calibration

In order to ensure a homogeneous time series of brightness temperatures from all channels which are available from the SMMR across the SSM/I to the SSMIS sensors, the varying individual instrument characteristics have to be corrected by an inter-sensor cross calibration of the individual radiometers.

An absolute inter-calibration of the microwave instruments is not possible because there are no absolute reference SI-traceable targets available. The first step for an inter-calibration is therefore to select a suitable reference target and then to transfer this calibration to all other sensors. This inter-sensor calibration method is designed to provide a homogeneous time series with any potential absolute, but unknown, offset observed in the reference brightness temperatures transferred to all other radiometers. The inter-calibration reference selected for the CM SAF FCDR is the SSM/I aboard DMSP F11. In tests with different wind speed algorithms against collocated in-situ buoy data this radiometer exhibits a reliable long-term stability. Furthermore, it has a temporal overlap with most of the other SSM/I radiometers which does allow a direct inter-calibration. Selecting a reference in the middle of a chain of instruments also minimises the number of inter-calibration steps needed to harmonise the full constellation. Each inter-calibration step introduces a new uncertainty. Selecting a reference at the end of the chain would therefore result in a large uncertainty at the begin of a climate data record.

### 5.1   SSM/I and SSMIS Earth incidence angle normalisation

The SSM/I scanning system is designed to observe the Earth's surface under a nominal constant Earth incidence angle (EIA) of $53°$. A deviation of one degree from the nominal incidence angle causes $T_B$ changes of up to 2 K depending on surface and atmospheric conditions. Errors in the geophysical parameters can be in the order of 5% to 10% if the variation in the incidence angle is not taken into account (Furhop and Simmer, 1996).

The incidence angle varies mainly due to changes in the altitude of the spacecraft, caused by the eccentricity of the orbit and the oblateness of the Earth. The EIA can also vary across the scan by approximately $0.1°$ due to pitch and roll variations of the spacecraft. For example, the altitude of the F13 varies along the orbit between 840 to 880 km, corresponding to a variation of about $0.3°$ in EIA. Also a seasonal variation is apparent with a period of about 122 days. This periodic change is caused by the precession of the argument of perigee due to the Earth's oblateness. The strongest variation is found for the F10 with about $1°$ due to a highly elliptical orbit. But not only the variation, also the mean EIA differs across the radiometers and even varies for different channels of an SSMIS because the individual feedhorns are slightly differently aligned.

The varying EIA must be taken into account either explicitly by a surface emissivity model in a geophysical retrieval scheme or by normalising the $T_B$s to a nominal EIA value. This is not only the case when geophysical parameters are derived but also an important prerequisite for the inter-sensor calibration procedure as two sensors observe the same area at the same time but at different EIAs. The normalisation is made available as an additional offset variable in the CM SAF FCDR data files, which can be added to the inter-calibrated $T_B$s. The offsets are computed using the method described by Furhop and Simmer (1996) for water type FOVs only. This method uses the observed $T_B$ vector to estimate the slope of the $T_B$ change with respect to the



incidence angle $\theta$:

$$sl = \frac{\delta T_B}{\delta \theta} \ . \tag{22}$$

The $T_B$s are then normalised in an iterative procedure:

$$sl = a_0 + \sum_i a_i \cdot T_B \left( \theta = 53^\circ \right)$$

$$T_B \left( \theta \right) = T_B \left( \theta = 53^\circ \right) + sl \cdot \Delta \theta \ , \tag{23}$$

where $a_i$ are the regression coefficients of the Furhop and Simmer model for the $T_B$s observed at 19, 22, and 37 GHz. The slope $sl$ depends on the atmospheric and surface conditions. The procedure is only applicable over ocean because the regression coefficients are determined for this surface type. The final normalisation offset $\Delta T_B^{eia}$ is then defined as:

$$\Delta T_B^{eia} = T_B \left( \theta = 53^\circ \right) - T_B \left( \theta \right) \ . \tag{24}$$

## 5.2 SSM/I and SSMIS inter-calibration model

The SSM/I and SSMIS sensors are spectrally inter-calibrated and thus providing a seamless data record of the SSM/I frequencies at 19, 22, 37, and 85 GHz. The SSM/Is aboard F08 and F15, which do not have a temporal overlap with F11, are calibrated to the corrected F10 and F13 radiometers, employed as transfer standards, respectively. Also the SSMIS sensors do not have a temporal overlap with F11. Here the SSMIS$_{F16}$ is inter-calibrated to the corrected SSM/I$_{F13}$ radiometer in a first step. Then the corrected SSMIS$_{F16}$ is used as the transfer standard for the SSMIS radiometers aboard F17 and F18. The inter-calibration model is an updated version of the homogenisation scheme described in Andersson et al. (2010). The largest contributions to the expected systematic uncertainties, and thus the sources of expected differences, arise from the calibration non-linearity and the imperfect knowledge of the radiometer cross-polarisation leakage and spillover loss (see Sect. 6). In order to account for these issues, the inter-calibration model was designed to correct for observed inter-sensor $T_B$ differences as a function of scene temperature and the scene polarisation difference $\Delta_{pq} = T_{Bv} - T_{Bh}$. The model comprises an offset term $a$, a scene dependent linear correction term $b$, a polarisation difference term $c$, and a non-linearity term $d$. The non-linearity correction depends on the temperature difference of the observed scene temperature from both calibration targets and is applied as a part of the radiometric calibration on the antenna temperatures:

$$T_A^{\#} = T_A + d \cdot \left( T_A - \left\langle T^h \right\rangle \right) \cdot \left( T_A - T^c \right) \ . \tag{25}$$

The other inter-calibration terms are applied after the antenna pattern corrections:

$$T_B^{\#} = APC(T_A^{\#})$$

$$T_B^{ic} = a + b \cdot T_B^{\#} + c \cdot \Delta_{pq} \tag{26}$$

The reason to split the inter-sensor calibration procedure is to keep the corrections as logical enhancements to specific processing steps in the calibration and the antenna pattern correction.





The individual inter-sensor calibration terms are determined by fitting the inter-calibration model Eq. (25) and Eq. (26) using match-up data sets of the radiometers compiled from the final $T_B$s on the lower SSMI(S) resolutions. This includes all corrections as described in Sect. 3 plus the incidence angle normalisation offset $\Delta T_B^{eia}$ (see Sect. 5.1) in order to derive the inter-calibration coefficients at a fixed nominal EIA.

The data sets used for the inter-calibration are constructed from absolute $T_B$s of oceanic, sea-ice and cold land scenes plus
the polarisation differences over all surface types. Each channel is binned into a daily global 1° equal angle grid, separately for morning and evening orbits. This selection of $T_B$s uses the Earth as an inter-calibration transfer target, with the relative cold ocean as a vicarious cold reference and the warm land (via the polarisation difference) as a vicarious warm reference inter-calibration target. This procedure also ensures that the complete natural variability of the observations is taken into account, and thus no extrapolation is required. For each radiometer a match-up data set with the inter-calibration reference or the selected
(and corrected) transfer standard is compiled from the collocated gridded mean normalised $T_B$s for the entire overlap period. Only grid cells with morning and evening (local time) observations are selected and averaged to daily mean observations in order to minimise the influence of diurnal cycle variations, assuming sinusoidal characteristics. The polarisation differences are much less affected by the diurnal cycle as the vertical and horizontal polarisation at the same frequency exhibit similar diurnal variations. These daily mean observations are finally averaged to produce monthly mean $T_B$ maps. The monthly mean 1° grid
cells are treated as individual independent observations and the calibration coefficients in Eq. (25) and Eq. (26) are fitted by minimising the observed temperature differences to the selected target instrument.

Due to the stable in-orbit calibration of the SSMI(S) sensors, the coefficients are considered to be constant during the lifetime of a radiometer and are used to homogenise the measurements of the different instruments. The inter-calibration offsets $\Delta T_B^{ic}$ to the $T_B$s are assumed to be valid over all surface types and computed during the final processing for each FOV and then
separately archived in the FCDR data files:

$$\Delta T_B^{ic} = T_B^{ic} - T_B \ . \tag{27}$$

By keeping the incidence angle normalisation offsets $\Delta T_B^{eia}$ and the inter-calibration offsets $\Delta T_B^{ic}$ separately from the $T_B$ values itself, each user has the freedom to choose whatever correction or combination is adequate for their application. The inter-calibration can be applied independently of the EIA correction with an error of $< 0.1$ K.

**5.3 SMMR inter-calibration model**

The overlap period of SMMR and SSM/I$_{F08}$ is just about one month. This is not a sufficient record length to directly derive stable inter-calibration coefficients as seasonal or other multi-monthly variations can not be accounted for. It is therefore not possible to use the statistical methods developed for the SSMI(S) inter-calibration. Instead, a double-difference technique using modelled brightness temperatures from reanalysis profiles is employed here following the approach from Sapiano et al. (2013).
It is important to keep in mind that this inter-calibration of the SMMR is not intended to completely remove all observed differences. Instead the measured SMMR $T_B$s are corrected to be physically consistent with corresponding SSM/I $T_B$s. This means small differences due to different nominal channel centre frequencies (e.g. 18 GHz versus 19 GHz) and different band-





widths are to be expected. Also the different nominal EIAs of the measurements ($50°$ versus $53°$) will result in overall and time-varying differences. These must not only be accounted for during the inter-calibration procedure in order to derive physi-
cally consistent inter-calibration coefficients, but also later on in the application. This can be done either directly in the retrieval algorithm or indirectly by normalising the observed $T_B$ to a nominal frequency and EIA.

Using a reanalysis as a transfer standard has advantages but also limitations. Firstly, the most important requirement here is the temporal stability of the reanalysis. This is the basic underlying assumption to allow a transfer of an observed bias from the SSM/I time period to the SMMR time period. Secondly, the uncertainties in the applied radiative transfer model and the
surface emissivity model should be well behaved in order to cancel out in the double difference technique.

While using modelled $T_B$s for the inter-calibration, the EIA dependence of the $T_B$s and the frequency shifts are explicitly accounted for. Also the different equator overpass times of Nimbus-7 (12 AM) and DMSP F08 (6 AM) are considered by the temporal collocation criterion. However, a limitation is the large uncertainty of surface emissivity models over land, ice and snow covered surfaces due to unknown surface emissivities. Also the strong diurnal cycle over land is not resolved by
the reanalysis time steps. Additionally, rainy and cloudy scenes increase the uncertainty of the radiative transfer model due to scattering effects of the water droplets and ice crystals. In order to minimise these influences, only cloud-free and lightly cloudy scenes over water surfaces are selected for the inter- calibration match-up data sets.

The profile and surface data used here are taken from the ERA-20C reanalysis (Poli et al., 2013). Brightness temperatures at the top of the atmosphere are calculated with RTTOV 11.2 (Saunders et al., 2013) and the surface emissivity with FASTEM 6
(Meunier et al., 2014). Profiles from ERA-20C are available every 3 hours, resulting in maximum time differences of 90 minutes between observed and simulated $T_B$s. The simulations are done for the entire time period covered by the non-intercalibrated SMMR and the SSM/I FCDR for F08. In order to reduce the uncertainty in the observed differences, the match-up data is gridded into daily global $1°$ equal angle grids, separately for ascending and descending orbits. As ERA-20C does not include satellite observations, this likely assures that the $T_B$s in both independent data records are stable over the selected time period.

No significant trends can be observed in the global monthly mean differences between simulated and observed $T_B$s for the SSM/I$_{F08}$ $T_B$s. Morning and evening SSM/I $T_B$ anomalies depict nearly identical temporal variations and agree to within 0.2 K in the mean value (not shown). Only the channel at 37 GHz,h-pol shows a larger difference with a value of 0.4 K between ascending and descending orbits. This also underlines the stability and quality of the SSM/I FCDR. The SMMR anomalies behave differently. Generally they are stable over time despite the last months of 1987 where the differences get
smaller in all channels. The 21 GHz,v-pol channel seems to be affected by a small upward drift towards the end of its lifetime (1983-1985). These trends are most likely caused by the correction of the long-term trend in the level 1B data record (see Sect. 3.6.8). As this correction was derived from a polynomial fit of the absolute global mean values, existing natural climate trends could have been not accounted for. Also seasonal variations of the ascending and descending orbits can be observed, which do not agree. Phase shifts in the anomalies are visible for the channels at 18 GHz,v-pol, 21 GHz,v-pol, and both channels
at 37 GHz. The amplitude of the variations decreases with increasing frequency from about 2 K to 0.5 K.

It was not possible to identify the root cause for these temporal variations and offsets. The magnitude of the observed $T_B$ anomalies and their overall decrease with frequency might be caused by unresolved variations in the EIA. This can be either





due to errors in the satellite attitude control or deficiencies in the applied short-term, orbit angle dependent, correction in the level 1B data record. These short term corrections were derived by forcing ascending minus descending $T_B$ differences to be
zero without accounting for the different EIA values at ascending and descending paths or diurnal variations. As this level 1B correction is based on look-up tables, which are not available, it can not be reverted. Hence, it is not possible to separate the effects and identify the physical reasons for the observed seasonal variations.

The inter-calibration model employed for the CM SAF SMMR data record is adapted from the absolute calibration correction as applied in the level 1B data processing. This correction is a scene dependent correction to a modelled background acting as a
cold vicarious calibration target while keeping the warm calibration target at the instrument warm load reference temperature. Selecting a simple $T_B$ inter-calibration offset derived at the cold end would result in scene dependent biases at warm scene temperatures. The cold target for the FCDR inter-calibration is now defined as the inter-calibrated SSM/I$_{F08}$ $T_B$ acting as a transfer target to the SSM/I$_{F11}$ reference sensor. Keeping the nomenclature from Eq. (17) (Njoku et al., 1998), a linear correction is then defined as:

$$T_B^{ic} = a + c \cdot T_B \,, \tag{28}$$

where $T_B$ is the observed SMMR brightness temperature, $T_B^{ic}$ is the inter-calibrated value and $a$ and $c$ are the inter-calibration coefficients. The constraints can be defined as:

$$T_B^c = T_B^o + \left(T_B^m - T_B^o\right) - \left(T_B^{\#m} - T_B^{\#o}\right)$$
$$T_B^c = a + c \cdot T_B^o$$
$$I' = a + c \cdot I' \,, \tag{29}$$

where $T_B^c$ is the mean corrected SMMR brightness temperature, $T_B^o$ is the mean observed SMMR brightness temperature and $T_B^m$ is the modelled brightness temperature. $\left(T_B^m - T_B^o\right)$ is the mean SMMR difference between modelled and observed temperatures, $\left(T_B^{\#m} - T_B^{\#o}\right)$ is the corresponding mean SSM/I brightness temperature difference and $I'$ is the normalised warm load reference brightness, which shall not change. These equations are solved for the inter-calibration coefficients $a$ and
$c$:

$$a = I' \cdot \frac{T_B^c - T_B^o}{I' - T_B^o}$$
$$c = \frac{I' - T_B^c}{I' - T_B^o} \tag{30}$$

The coefficients $a$ and $c$ depend on the actual warm load equivalent brightness $I'$ and are updated online for each new set of calibration data. The inter-calibration offsets $\Delta T^{ic}$ to the observed $T_B$s are assumed to be valid over all surface types and
computed during the final processing for each FOV and then archived in the SMMR FCDR data files.





## 6 Uncertainty estimates

It is not possible to specify a complete systematic uncertainty estimate for the observed $T_B$s because a general limitation to trace observed systematic differences back to SI standards for microwave observations (Hollinger et al., 1990). As there are no absolute reference observations available to validate the absolute radiometer calibration, each instrument must be treated as a

unique observing system. This means the measured brightness temperatures can only be compared against similar observations, for example aircraft measurements with an SSM/I or SSMIS simulator, or against simulated $T_B$s computed with a radiative transfer model. In this case, the exact channel specifications must be known. But also the radiative transfer model and the employed surface emissivity model will have random and systematic uncertainty sources, that have to be properly characterised. The observed and the modelled $T_B$s can then be compared to test whether they agree within their defined total uncertainties.

Both methods have been used to evaluate the absolute calibration of the SSM/I during the F08 instrument cal/val program (Hollinger et al., 1990). The conclusion from these evaluation experiments is that the absolute calibration shows no significant systematic bias and may be better than the uncertainty of its determination of $\pm 3$ K.

The total expected uncertainty of a quantity can be divided into two components: a systematic uncertainty and a random uncertainty. While the random uncertainty of the $T_B$s can be determined on-orbit statistically from the instrument measurements

used to calibrate the radiometer, the total systematic uncertainty must be identified during pre-launch ground measurements, following the ISO Guide to the expression of uncertainties (ISO GUM, JCGM 100:2008).

### 6.1 Random uncertainty

The random uncertainty of a radiometer is usually expressed as a noise equivalent temperature Ne$\Delta$T, which is the standard uncertainty of an individual measurement. This uncertainty is defined as the standard deviation of the measurement referenced

to the energy of the radiation incident on the antenna. This noise can be estimated using the on-orbit calibration measurements and the calibration equation. Applying standard error propagation to the SSMI(S) radiometric calibration Eq. (2) and Eq. (3) and dropping the averaging symbols here to increase readability:

$$\text{Ne}\Delta\text{T}^2 = U\left(T^h\right)^2 + U\left(C^h\right)^2 + U\left(C^c\right)^2 + U\left(C^e\right)^2 \tag{31}$$

with the following standard uncertainties contributing to the combined total standard variability:

(a) Standard uncertainty due to noise in the smoothed warm target temperature readings $\sigma_{T^h}$:

$$U\left(T^h\right) = \frac{\delta T_A}{\delta T^h} \cdot \sigma_{T^h} = \frac{C^e - C^c}{C^h - C^c} \cdot \sigma_{T^h} \, ,$$

(b) Standard uncertainty due to noise in the smoothed warm calibration counts $\sigma_{C^h}$:

$$U\left(C^h\right) = \frac{\delta T_A}{\delta C^h} \cdot \sigma_{C^h} = \frac{\left(T^h - T^c\right) \cdot \left(C^c - C^e\right)}{\left(C^h - C^c\right)^2} \cdot \sigma_{C^h} \, ,$$

(c) Standard uncertainty due to noise in the smoothed cold calibration counts $\sigma_{C^c}$:

$$U\left(C^c\right) = \frac{\delta T_A}{\delta C^c} \cdot \sigma_{C^c} = \frac{\left(T^h - T^c\right) \cdot \left(C^e - C^h\right)}{\left(C^h - C^c\right)^2} \cdot \sigma_{C^c} \, ,$$



(d) Standard uncertainty due to noise in the Earth counts $\sigma_{C^e}$:

$$U\left(C^e\right) = \frac{\delta T_A}{\delta C^e} \cdot \sigma_{C^e} = \frac{T^h - T^c}{C^h - C^c} \cdot \sigma_{C^e} \,.$$

The standard deviations of the smoothed cold counts $\sigma_{C^c}$, smoothed warm counts $\sigma_{C^h}$ and smoothed temperature readings $\sigma_{T^h}$ are estimated from all available valid measurements on a daily basis. The daily mean noise equivalent temperature Ne$\Delta$T

for each channel can then be estimated while viewing the warm target, i.e. replacing $C^e$ with $C^h$, and applying the standard deviation of the individual warm counts. As the variability of the warm counts is generally larger than the variability of the cold counts this value is a valid estimate for the maximum expected Ne$\Delta$T. Due to bandwidth limitations, the cold and warm load readings of the SSMIS instruments are only available as a mean value over a pre-defined number of beam positions (scan-line mean). This number differs between the instruments and also changes over time. The variance of the individual readings are

therefore estimated from the variance of the scan-line mean values by multiplying these with the number of averaged beam positions valid at that observation time.

The term with the largest contribution to the total standard uncertainty is the uncertainty due to noise in the Earth counts $U\left(C^e\right)$ with about 98%. The contribution from the uncertainty in the temperature readings can be neglected while the uncertainty in the cold and warm calibration target readings contribute the remaining 2%. The across-scan smoothing applied to the

scanline mean values Eq. (5) reduces Ne$\Delta$T by approximately 10%. While reducing the noise of the individual observations, the across-scan averaging introduces a small error covariance over the smoothing kernel width.

The Ne$\Delta$T for the SMMR is derived from the corresponding radiometric calibration Eq. (7):

$$
\begin{aligned}
\text{Ne}\Delta\text{T}^2 &= U\left(I\right)^2 + U\left(S\right)^2 + U\left(C^h\right)^2 \\
&\quad + U\left(C^c\right)^2 + U\left(C^e\right)^2
\end{aligned}
\tag{32}
$$

with the following standard uncertainties contributing to the combined standard variability:

(a) Standard uncertainty due to noise in the calibration brightness $I$:

$$U\left(I\right) = \frac{\delta T_A}{\delta I} \cdot \sigma_I = S \cdot \frac{C^e - C^h}{C^c - C^h} \cdot \sigma_I \,,$$

(b) Standard uncertainty due to noise in the calibration slope coefficient $S$:

$$U\left(S\right) = \frac{\delta T_A}{\delta S} \cdot \sigma_S = \frac{C^e - C^h}{C^c - C^h} \cdot \sigma_S \,,$$

(c) Standard uncertainty due to noise in the averaged warm counts $C^h$:

$$U\left(C^h\right) = \frac{\delta T_A}{\delta C^h} \cdot \sigma_{C^h} = S \cdot \frac{C^e - C^c}{\left(C^c - C^h\right)^2} \cdot \sigma_{C^h} \,,$$

(d) Standard uncertainty due to noise in the averaged cold counts $C^c$:

$$U\left(C^c\right) = \frac{\delta T_A}{\delta C^c} \cdot \sigma_{C^c} = S \cdot \frac{C^h - C^e}{\left(C^c - C^h\right)^2} \cdot \sigma_{C^c} \,,$$



(e) Standard uncertainty due to noise in the earth counts $C^e$:

$$U\left(C^e\right) = \frac{\delta T_A}{\delta C^e} \cdot \sigma_{C^e} = S \cdot \frac{1}{C^c - C^h} \cdot \sigma_{C^e} \,,$$

The contributing standard deviations are estimated on a daily basis similar to the SSMI(S) but using the cold target because the variability of the SMMR cold counts is larger than the variability of the warm count. All derived variances and NeΔT values are archived in the FCDR data files as daily mean values in order to provide full traceability.

## 6.2 Systematic uncertainty

The total systematic radiometer calibration uncertainty budget can be composed of the following contributors: (a) warm load reference uncertainty, (b) cosmic background reference uncertainty, (c) radiometer/calibration non-linearity, (d) radiative coupling $\varepsilon$ uncertainty, (e) uncertainty in the APC coefficients, (f) reflector emissivity, and (g) along-scan cross polarisation mixing. Evaluation results for the SSM/I radiometric calibration have been reported by Hollinger et al. (1989) and Colton and Poe (1999), for the SSMIS by Kunkee et al. (2008c) and for the SMMR by Njoku et al. (1998). These results are discussed in the following sections and uncertainty values for each contributing term are estimated. The derived uncertainties are summarised in Table 1.

### (a) Warm load reference uncertainty

The calibration uncertainty of the warm load was determined pre-launch for the SSM/I$_{F08}$ by comparison with a variable precision calibration reference target over a range of 100 K to 375 K during thermal vacuum calibration. These tests show no systematic calibration error (Hollinger et al., 1989). Kunkee et al. (2008c) refer to an absolute uncertainty of 0.1 K for the SSMIS. However, no information on pre-launch measurements of the temperature sensors is available for the SMMR. A common value of 0.1 K is employed for all instruments.

### (b) Cosmic background reference uncertainty

An analysis of the calibration reflector antenna patterns, when the SSM/I is in the calibration position, reveals that the reception of Earth and spacecraft radiation is extremely small (less than a few tenth of a degree). Thus it is believed that the SSM/I calibration reflector provides a clear view of the cosmic background to the feedhorn and hence provides a highly accurate blackbody calibration reference (Hollinger et al., 1989). However, this is not the case during phases when parts of the moon are in the field of view of the reflector (see Sect. 3.6.2). This short-lived intrusion events can cause an underestimation of the true scene temperature of up to 0.5 K, depending on the channel frequency and scene temperature. As the SSMIS employs the same cold calibration method, the same problem arises and the same maximum bias is assumed.

The SMMR suffers from periodic sunlight intrusion events into the cold sky calibration horns. This usually occurs once during each orbit over the southern polar region. These regions are not used in the calibration. Instead the cold counts are linearly interpolated between the last valid values (Njoku et al., 1998). Apart from these events, it is believed that the SMMR calibration horns provide a clear view of the cosmic background.




To account for these short-term SSMI(S) intrusions and the uncertainties in the SMMR calibration interpolation a mean uncertainty of 0.1 K is assumed for all instruments, keeping in mind that only small parts of the orbit are affected.

**(c) Radiometer/calibration non-linearity**

Non-linearities in the calibration measurements and in the radiometer receiver may be expected. This can impact the radiometric calibration when a two-point linear calibration equation is applied. At observation temperatures equal to the calibration
reference targets, the calibration uncertainty is the accuracy of the reference, which does not show a systematic uncertainty (see above). At intermediate temperatures, radiometer non-linearity and calibration reference temperature uncertainty contribute to the total uncertainty with the uncertainty weighted according to the difference between the observed scene and the calibration references. To test the radiometer absolute calibration, preflight thermal vacuum tests have been conducted for the SSM/I (Colton and Poe, 1999). Calibration cycles were run for three different sensor temperatures: cold ($0°C$), ambient ($28°C$) and
hot ($38°C$). For the on-orbit expected operating range the calibration errors are typically -1.4 to 0.5 K (Colton and Poe, 1999) and hence within the instrument specifications. The corresponding graphs in Colton and Poe (1999) point to a scene temperature dependent bias with a larger underestimation for colder scene temperatures. However, the measurements are believed to contain artefacts due to the actual test setup. The on-orbit calibration are expected to perform better than the pre-launch calibration tests indicate (Colton and Poe, 1999).
To test the SSMIS radiometer absolute calibration, thermal vacuum tests have been conducted for all SSMIS instruments, with all meeting the accuracy requirement of 1 K (Kunkee et al., 2008b). However, a scene temperature dependent offset is depicted in Fig. 6a in Kunkee et al. (2008b). Without more details being available, the same values as for the SSM/I are used.

    Thermal vacuum tests have also been conducted for the SMMR instrument at the Jet Propulsion Laboratory (JPL) for temperature levels ranging from 77 K to 320 K (Gloersen, 1987). Within the observed radiometer switch temperature range of
$\pm 2$ K the differences between linear and quadratic calibration equations are found to be within 0.1 K for all channels.

**(d) Radiative coupling $\varepsilon$ uncertainty**

The radiative coupling coefficient $\varepsilon$ was originally determined for the SSM/I$_{F08}$ and was used unchanged in the computation of the antenna temperatures Eq. (6) since then. The correction due to this factor depends on the temperature difference between the warm load and the forward radiator plate. This correction is typically in the order of 0.1 K for an expected temperature
difference of 10 K. However, some instruments show a very strong seasonal variation in the temperature difference reaching 40 K. A relative uncertainty of 1% in $\varepsilon$ can then lead to a bias of 0.4 K in the scene brightness temperature.

**(e) Uncertainty in the APC coefficients**

During the ground instrument calibration also a radiometric characterisation of the antenna properties is performed. The performance was established from analyses of the antenna gain function as measured on an antenna range from which, amongst oth-
ers, feedhorn spillover and cross-polarisation loss were determined. The estimated absolute accuracy of the feedhorn spillover





is 0.3-0.5% and the relative cross-polarisation accuracy is in the order of 5-10% (Colton and Poe, 1999). The uncertainty in the spillover translates to a scene dependent uncertainty in the order of about 1-1.5 K at 300 K scene temperature. The uncertainty in the cross-polarisation accuracy results in an uncertainty, depending on the scene polarisation difference, in the order of 0.15-0.3 K for a scene with 50 K difference between vertical and horizontal polarisation. These values are also employed for
the SMMR and SSMIS.

**(f) SSMIS reflector emissivity**

The emissive reflector problem is specific for the SSMIS. It leads to an additional systematic uncertainty which depends on the emissivity of the reflector (see Sect. 3.6.7). The emissivity itself is a function of the channel frequency and the temperature difference between the reflector and the observed target. Thus it is not only scene dependent but also a function of the orbit
position when the reflector is heated by the solar radiation and cools in the Earth shadow. The strongest errors are to be expected when the spacecraft leaves the Earth's shadow with the reflector temperature increasing by about 80 K and at low scene temperatures. With the reflector temperature reaching about 300 K and lowest scene temperatures around 150 K a maximum difference of about 150 K can be expected. The error due to the emissivity of the reflector can then reach about 13 K for some of the sounding channels with an estimated emissivity of $\varepsilon_{refl} = 0.09$. The largest error for the SSM/I like
environmental and imager channels appears at 91 GHz for $\text{SSMIS}_{F17}$. With an emissivity of 0.04 at this frequency, this can result in an error of about 6 K. This emissivity induced error is corrected in the CM SAF FCDR. The expected uncertainty in the determination of the emissivity can be estimated as $\pm 10\%$, which then translates to a maximum remaining error of $\pm 0.6$ K.

**(g) Along-scan cross polarisation mixing**

This systematic uncertainty source is specific for the SMMR as a result of the instruments design. The antenna system consists
of a scanning parabolic reflector and a fixed multispectral feedhorn. This leads to a scan-angle dependent polarisation mixing, which is corrected for when the antenna temperature is converted to the $T_B$. This method has been developed after launch by Gloersen (1987). Njoku et al. (1998) adopted this method and report the uncertainty of the fitted correction coefficients in the Level 1B data record to be between 0.046 K and 0.157 K with the remaining along-scan biases range between 0.2 K to 0.5 K.

**Total systematic uncertainty**

The individual systematic uncertainty contributions are summarised in Table 1. If only a maximum range can be estimated, the corresponding standard uncertainty $u_i$ is derived from the uncertainty range $y_i$ assuming a uniform distribution of the uncertainty within the expected range of variation $\Delta a_i = \left| \frac{y_+ - y_-}{2} \right|$:

$$u_i(y) = \frac{\Delta a_i}{\sqrt{3}} \tag{33}$$





The combined total expected standard systematic uncertainty

$$u_c(T_B) = \sqrt{\sum_i u_i^2(y)} \tag{34}$$

is then in the order of 0.7-1.1 K for the SSMI(S) and 0.4-1 K for the SMMR, with a significant dependence on the scene temperature due to the non-linearity and spillover terms and a minor dependence on the scene polarisation. Overall, the total expected systematic uncertainty of the absolute calibration depends on the frequency, the scene temperature, and the polarisation difference. As the random uncertainty scales with the number of observations, the systematic uncertainty has to be accounted for when comparing two climatological means of $T_B$s from different platforms.

## 7 Evaluation

A full evaluation of the CM SAF FCDR was done and the detailed results are presented in the corresponding validation report, which is freely available from the DOI landing page https://doi.org/10.5676/EUM_SAF_CM/FCDR_ MWI/V003. The corrected and inter-calibrated $T_B$s were compared to the uncorrected raw data records (RDR) and to another SSMI(S) brightness temperature data record from Kummerow et al. (2013). The homogeneity of the data records is tested by comparing against the respective ensemble mean of the available satellites in each data record and additional statistical values are given for bias, robust standard deviation (RSD), median absolute deviation (MAD) and decadal stability.

Conducting a direct inter-sensor validation for the SMMR, as it was done for the SSMI(S) instruments, is not feasible. The overlap time period is just a month and SMMR data is available just every other day. In case of the 21 GHz channels, there is no overlap at all as these channels failed in 1985. Also the channels are not at the exact same centre frequency (18 and 21 GHz instead of 19 and 22 GHz) and the SMMR inter-calibration does not account for a frequency shift. Therefore a common stable reference has to be used for both instruments and the anomaly against this reference needs to be analysed.

### 7.1 Inter-sensor differences between SSM/I and SSMIS

In order to quantify the consistency of the brightness temperatures across the SSMI(S) sensors, a reference has to be established first. As there is no absolute reference available and the operating sensors change over time, the ensemble mean of all available instruments at each month has been selected as the relative reference. This approach simplifies the further analysis, as it can be performed for each sensor. The inter-sensor differences are derived by comparing the respective bias values to the ensemble mean. The maximum inter-sensor difference is the ensemble spread, which characterises the observation uncertainty.

A global monthly mean bias is the mean systematic offset to another sensor. However, it is also important to quantify the observed regional differences, which are characterised by the dispersion of all gridded monthly mean $T_B$ differences around the global mean. The median of absolute differences ($MAD$), without correcting for the mean systematic offset ($BIAS$), is a measure of the total absolute uncertainty. A robust (resilient to outliers) measure of the statistical dispersion of a distribution is the median absolute deviation about the median. Assuming a normal distribution, the expected $RSD$ can be estimated from the median absolute deviation by scaling it with a factor of $1.48$.




For the evaluation the data record has been gridded to equal angle 1° monthly mean global fields separately for AM and PM orbits. For this comparison all oceanic and sea-ice covered grid cells are used in order to minimise diurnal cycle variations. To derive the relative instrument differences, an ensemble mean data set has been compiled on a monthly basis for all instruments and channels. The ensemble monthly mean grid box brightness temperature $\langle T_B \rangle$ for each month $t$ and grid box index $g$ is calculated as the arithmetic mean of the individually gridded monthly mean brightness temperatures $T_B$ from all available

instruments $s$:

$$\langle T_B \rangle (t,g) = \frac{1}{N_s} \cdot \sum_{i=1}^{N_s} T_B (s = si, t, g) \, , \tag{35}$$

with $N_S$ as the number of contributing instruments for each grid box and month. The distribution of brightness temperature differences $\Delta T_B$ relative to this ensemble mean

$$\Delta T_B (s,t,g) = T_B (s,t,g) - \langle T_B \rangle (t,g) \tag{36}$$

is then statistically analysed. Robust statistics are applied, with $M$ as the median of $\Delta T_B$ for each instrument $s$. The following statistical quantities are analysed:

$$BIAS_{si} = M \left[ \Delta T_B (t,g,s=si) \right] \tag{37}$$

$$MAD_{si} = M \left[ |\Delta T_B (t,g,s=si)| \right] \tag{38}$$

$$RSD_{si} = M \left[ |BIAS_s - \Delta T_B (t,g,s=si)| \right] \cdot 1.48 \tag{39}$$

The inter-sensor bias is derived as the difference between two sensors $(BIAS_{s1} - BIAS_{s2})$.

    The decadal stability $t_D$ for each channel and instrument is estimated using a linear model trend, fitted to the time series of monthly anomalies relative to the ensemble mean. The monthly anomalies are defined as the median of the global distribution of differences $\Delta T_B$ at each monthly time step $t$:

$$\overline{\Delta T_{B,si}} (t) = M \left[ \Delta T_B (g, s=si) \right] (t) \, . \tag{40}$$

A simple model with a linear decadal trend $t_D$ in K/decade can then be defined as:

$$\overline{\Delta T_{B,si}} (t) = \overline{\Delta T_{B,si}} (t=0) + \frac{t_D}{120} \cdot t + \varepsilon(t), \tag{41}$$

with $\varepsilon(t)$ representing the fraction of the monthly anomalies not explained by the linear approximation. The linear model terms (offset and trend) are found by a least square regression fit. The results of the statistical analysis for all channels are shown in detail in the validation report. An example for the channel at 37 GHz,h-pol is presented in Fig. 8. The top image

shows $T_B$ anomalies of the RDR without any modification or correction. The second plot depicts the $T_B$ anomalies after the corresponding corrections (reflector emissivity correction, sunlight intrusion, moonlight intrusion) and normalisation (EIA normalisation and diurnal cycle) are applied. The third graph shows the final $T_B$ anomalies after the inter-calibration.

    The time series of the raw data records show a very diverse picture. The agreement in the raw data records between the SSM/I is generally better than between the SSMIS. This is also evident for the homogenised data records that have been corrected for





reflector emissivity, incidence angle and diurnal cycle variations (second panel). Most $T_B$ differences for the SSM/I instruments are below 0.5 K in the RDR. The largest difference found for the SSM/I is about 1 K in the 85 GHz channels (not shown). In contrast to this, most of SSMIS differences are between 0.5 K to 1 K. The ensemble spread between the SSM/I and SSMIS (2006-2009) is larger than 3 K for the 37 GHz channels. Best agreement between both sensor types is found for the 19 GHz channels (not shown) with about 1 K difference. This means, most of the observed inter-sensor differences before applying the

inter-calibration offsets are larger than the estimated standard uncertainty of the SSMI(S) instruments, which is about 0.7 to 1.1 K.

The calibration enhancement due to intrusion and reflector emissivity corrections (from first to second panel) and the inter-calibration (from second to third panel) reduce the seasonal variability and the inter-sensor variations and increase the quality and stability of the SSMI(S) data record significantly. The global climatological mean inter-satellite bias between SSM/I and

SSMIS has been reduced to below 0.05 K. As for the raw data record, the agreement in the final FCDRs between the individual SSM/I instruments is generally better than between the SSMIS instruments.

The FCDR significantly reduces the observed differences between the monthly means and show very similar results for individual satellites. All global monthly mean inter-sensor differences are within 1 K. Comparing the absolute maximum difference between the individual satellite biases to the ensemble mean, the CM SAF FCDR shows a remaining ensemble

spread between 0.05 to 0.15 K. No significant trend can be observed.

Overall, the monthly anomalies show a variability which is larger for horizontally than for vertically polarised channels and increases with higher frequencies. This variability is caused by the inclusion of all scenes, i.e. no rain filtering is applied. This additional noise can be interpreted as an additional uncertainty due to differences in space and time collocation and sampling variability.

The robust standard deviations (not shown) depict constant values over time for all channels and satellites between 0.5 K and 1 K. This means, that about 70% of all analysed monthly mean grid boxes are within $\pm 0.5$ K to $\pm 1$ K, respectively.

Seasonal variations of the $T_B$ differences at 91 GHz, caused by the reflector emissivity issue affecting the instruments on-board F16 and F17, are in the order of 1 K for the global mean. These anomalies are corrected for the FCDR and only small variations remain in the homogenised plots (not shown).

## 7.2 Double differences

As mentioned before, an inter-sensor validation for the SMMR was not conducted, instead double differences between observed and modelled $T_B$s are analysed. Ideally one would select one reference for the inter-calibration and another one for the evaluation. However, it was decided to use only ERA-20C (Poli et al., 2013) because it relies only on surface pressure and sea-ice information and does not assimilate satellite data. Thus, ERA-20C is independent from the FCDR and also assumed to

be more stable than ERA-Interim.

In order to minimise the influence of diurnal cycle, unknown surface emissivities and scattering effects, the comparison was limited to cloud-free scenes and scenes with small liquid water content over water surfaces. Brightness temperatures at the top of the atmosphere are calculated for the filtered data record as described in Sect. 5.3. In order to reduce the uncertainty in the





observed differences, the match-up data is then gridded into daily global 1° equal angle grids, separately for ascending and
descending orbits and these are combined to daily means.

The monthly mean anomalies between the observed $T_B$ and the modelled $T_{BM}$ are calculated for each grid cell. Following
the notation defined in the previous section, the anomalies at each grid point are then defined as:

$$\Delta T_B (t, g, s) = T_B (t, g, s) - T_{BM} (t, g, s) \ . \tag{42}$$

To compare the characteristics of the time series for both instruments, global median $\overline{\Delta T_{B,s}} (t)$ and robust standard deviation
$RSD_s (t)$ are derived for each month.

Figure 9 shows the global monthly mean anomaly time series of SMMR and SSM/I$_{F08}$ for the channels at 37 GHz before
and after inter-calibration. The SSM/I$_{F08}$ itself is inter-calibrated to the SSM/I aboard F11, which is the FCDR calibration
reference instrument. This means the SSM/I$_{F08}$ is used as a transfer target. The relative differences without inter-calibration
range between -0.5 K and 5.3 K for the channels at 37 GHz,h-pol and 37 GHz,v-pol, respectively. Overall the SMMR anomalies
depict a larger seasonal variation compared to the SSM/I for the 19 and the 22 GHz channel (not shown here). After applying
the inter-calibration, all mean relative differences are significantly reduced to below 0.1 K. The time series for the 37 GHz
channels are stable and a good continuity to the SSM/I$_{F08}$ is apparent. The 21 GHz channel depicts an upward trend of about
1 K from 1984 until it failed in March 1985 (not shown here). The 19 GHz channels are showing a step in 1984 (not shown
here). This corresponds to an observed EIA change in 1984 and might be an indication of unresolved issues with the attitude
of the platform or some shortcoming in the surface emissivity model at that zenith angle. This is not evident in the 37 GHz
channels because the EIA dependence is smaller for higher frequencies. Also the seasonal variability is significantly larger for
the lower frequencies, compared to the SSM/I.

The anomalies are statistically analysed following the notations defined in the previous section. Recalling that about 70% of
all data samples are within one standard deviation of a Gaussian distribution, the mean robust standard deviation is a statistical
measure for the dispersion of all monthly mean 1° anomalies. However, it must also be accounted for the natural variability
and uncertainties in the forward model and the reanalysis data. Therefore only the relative differences between SSM/I$_{F08}$ and
SMMR are important, not the absolute values. The SSM/I$_{F08}$ is inter-calibrated to the reference instrument SSM/I$_{F11}$ and
the reanalysis is only used as a transfer standard. Under these conditions, all relative differences are within 0.1 K and are
not statistically significant. The $RSD$ of the SMMR horizontally polarised channels are similar to the SSM/I but the vertically
polarised channels show larger $RSD$s. Especially the channel at 22 GHz,v-pol depicts a significantly larger noise (now shown).

### 7.3  Comparison against reanalysis

Additionally to the inter-sensor comparison, a more complete analysis of each individual channel across the sensors for the
complete time period was conducted. The idea is to derive a better characterisation of the long-term FCDR stability, using
independent reanalysis data records. However, this approach is hampered due to several constraints. The basic assumptions are
that the reanalysis is independent of the observations, stable in time and does cover the complete time period of the FCDR.





Moreover, the uncertainties in the applied radiative transfer model and the surface emissivity model should be well behaved and characterised. However, there is actually no reanalyses available fulfilling all these constraints.

The FCDR is compared to the ERA-20C reanalysis, similar to the SMMR inter-calibration and evaluation, as ERA-20C does not assimilate any satellite observations. Note that the covered time period ends already in 2010. This does not allow a complete

analysis of the transition from SSM/I to SSMIS with F18 data becoming available in 2010. Comparisons to the ERA-Interim reanalysis (Dee et al., 2011) would cover the full temporal coverage of the FCDR. However, ERA-Interim assimilates SSM/I and SSMIS data records and is thus not independent. It was further shown, that the usage of different SSM/I instruments leads to inhomogeneities in the analysed water vapour from ERA-Interim (e.g. Schröder et al. (2016)). Thus, only results from the comparison to ERA-20C are shown here.

A serious limitation in the comparison is the large uncertainty of surface emissivity models over land, ice and snow covered surfaces due to unknown surface emissivities. Also the strong diurnal cycle over land is likely not fully resolved by the reanalysis time steps. Additionally, rainy and cloudy scenes increase the uncertainty of the radiative transfer model due to scattering effects of the water or ice droplets. In order to minimise these influences, only cloud-free and lightly cloudy scenes (small liquid water content) over water surfaces are selected for the match-up data sets. However, this practically limits this

comparison to the cold end of the natural spectrum and no conclusion can be drawn for the scene dependence or the warm end of the spectrum.

Brightness temperatures are modelled using the same method as in the previous section and also the statistical analysis is similar to the SMMR evaluation. The time series for AM/PM monthly mean anomalies and the $RSD$ are shown exemplarily for the channel at 37 GHz,h-pol in Fig. 10. The figure contains three images with time series of monthly mean values of anomalies

between RDRs (with reflector emissivity correction and intrusion correction applied) and ERA-20C (top), anomalies between inter-calibrated data records and ERA-20C (middle) and the $RSD$ of the inter-calibrated data record and ERA-20C (bottom).

The general conclusions from these comparisons are very similar to the results from the inter-sensor comparisons. The largest inter-sensor differences in the uncorrected data record are observed at the higher frequencies (37 GHz and 85 GHz) with values in the order of 3 to 4 K. Best agreement between the sensors is found for the 19 GHz channels (not shown) with values below

1 K. The applied inter-calibration effectively removes the observed differences. The residual $T_B$ difference is in the order of 0.1 to 0.2 K after applying the inter-calibration for most of the channels. This agrees with the maximum observed inter-satellite differences. A prominent feature is the application of a relative inter-calibration to the reference instrument aboard F11. From the model based evaluation it becomes clear, that the observed offset in the F11 is transferred to the all other instruments. This might not always be the optimum, but leads to homogeneous time series with all instruments acting as a synthetic F11. It is

not possible to see this in the inter-satellite comparisons were only relative differences are analysed. Conclusions concerning the temporal stability are difficult to derive. The anomalies from ERA-20C are quite stable over time but exhibit a large month-to-month variability. However, a small decrease in the mean anomaly over time can be observed in all channels. The individual FCDR instruments are independent and the inter-calibration is not time dependent, but all are depicting the same trend. The decrease is therefore assumed to be caused by changes in ERA-20C, possibly due to an improvement in the density

of assimilated data.



In order to get a linear trend analysis for the full SSM/I and SSMIS period, all individual instrument differences are averaged to a single data record for a specific frequency. The trend is then estimated using these mean anomalies. The results are consistent for all channels and all linear trends are between $-0.1$ to $-0.2$ K/decade. This is the same magnitude as the standard deviation of the global anomaly mean time series and therefore not significant. As mentioned before, most of this trend is most likely caused by ERA-20C.

## 8 Conclusions

A detailed description of the CM SAF FCDR of microwave imager radiances from SMMR, SSM/I and SSMIS is presented. This data record provides a consistent time series of brightness temperatures from October 1978 to December 2015. The inter-calibration method developed for the CM SAF FCDR explicitly includes all possible surface types to account for the entire natural distribution of brightness temperatures from radiometric cold scenes (rain-free ocean) to radiometric warm scenes (vegetated land surfaces). The inter-calibration offsets are determined for all SSM/I-like channels. Other channels on the SSMIS, not available on the SSM/I, are also re-processed, quality controlled, geolocated and included in the FCDR data files. The consistency and homogeneity of the FCDR was statistically analysed to demonstrate the improvement of the re-processed data records. The stability and homogeneity has also been tested by comparing the observed brightness temperatures against modelled brightness temperatures using reanalysis data.

The observed differences in the RDR range between 0.5 K to 3 K, depending on channel and instrument. Generally the differences are smaller within instruments from one series as compared to the inter-sensor difference across the SSMIS and SSM/I family. The overall mean differences in the CM SAF FCDR between the different sensors have been reduced to below 0.1 K, which is a significant improvement over the RDR. The mean $RSD$ for all channels and instruments has been significantly reduced. The observed residual variability in the inter-calibrated $T_B$s is mainly caused by the natural variability due to overpass time differences and sampling differences. No significant trend can be observed.

The SMMR has been evaluated using ERA-20C as a transfer standard. It was shown, that the global mean anomalies of SMMR and the SSM/I$_{F08}$ agree well after inter-calibration. The SMMR depicts higher values for the $RSD$ than the SSM/I, but still at the same magnitude. This is mainly due to higher NEdT values and poorer coverage with a 50% duty cycle.

Comparing against modelled brightness temperatures confirmed the results from the inter-sensor differences. However, it is still difficult to interpret the results in terms of temporal stability, as also the reanalysis might not be stable.

Finally it can be concluded that this FCDR is a fully documented and traceable data record providing greatly improved quality SMMR and SSMI(S) brightness temperatures as compared to original raw data records. It seamlessly extends the SSM/I data record with SSMIS data until end of 2015 and continues it with SMMR data to 1978. The final combined FCDR thus provides inter-calibrated brightness temperatures for the time period from 1978 to 2015. The SSM/I part of this FCDR has already been used by the ERA5 reanalysis (Hersbach et al., 2018).

It is planned to regularly update this FCDR within following Continuous Development and Operations Phases (CDOP) of the CM SAF in order to extend the covered time period with recent SSMIS data and to further improve the quality by integrating





user feedback. It is, amongst others, envisaged to improve the uncertainty characterisation and also to look at new sensors like
MWI and AMSR-3 for future usage.

## 9 Data availability

The full data record is freely available from the DOI landing page https://doi.org/10.5676/EUM_SAF_CM/FCDR_ MWI/V003
(Fennig et al., 2017). Also available from this webpage are an Algorithm Theoretical Basis Documents (ATBD), a Product User
Manual (PUM) and a Validation Report.

**Appendix A: Computation of brightness temperatures**

### A1 SSM/I

For the SSM/I, the variation of the polarisation angle over the main lobe (FOV) can be neglected and the Earth antenna
temperature equation is reduced to:

$$T'_{A_p} = \frac{1}{1 - \Lambda_{sp}} \int\limits_{\Omega_e} \left[ G_{pp} T_{B_p} + G_{pq} T_{B_q} \right] d\Omega \; . \tag{A1}$$

Following Ashcroft and Wentz (2000), the cross-polarisation leakage $\chi_p$ is defined as:

$$\chi_p = \frac{1}{1 - \Lambda_{sp}} \int\limits_{\Omega_e} G_{pq} d\Omega \; . \tag{A2}$$

With the assumption that the general cross-polarisation and co-polarisation gain shapes are similar:

$$G_{pq} \approx \frac{\chi_p}{1 - \chi_p} \cdot G_{pp} \; , \tag{A3}$$

this integral can be approximated as:

$$T'_{A_p} = \frac{1}{1 - \Lambda_{sp}} \int\limits_{\Omega_e} \left[ G_{pp} \left( T_{B_p} + \frac{\chi_p}{1 - \chi_p} T_{B_q} \right) \right] d\Omega \; . \tag{A4}$$

Also the antenna gains for vertical and horizontal polarisation are nearly identical $G_{pp} \approx G_{qq}$. The integral over the co-
polarisation gain can be substituted as the mean $T_B$ at polarisation $p$:

$$\overline{T}_{B_p} = \frac{1}{1 - \chi_p} \cdot \frac{1}{1 - \Lambda_{sp}} \cdot \int\limits_{\Omega_e} G_{pp} T_{B_p} d\Omega \tag{A5}$$

Equation (A4) can then be approximated as:

$$T'_{A_v} = (1 - \chi_v)\overline{T}_{B_v} + \chi_v \overline{T}_{B_h}$$
$$T'_{A_h} = (1 - \chi_h)\overline{T}_{B_h} + \chi_h \overline{T}_{B_v} \; . \tag{A6}$$



These two equations are solved for the scene $T_B$, which yields:

$$\overline{T}_{B_v} = T'_{A_v} + \frac{\chi_v}{1 - \chi_v - \chi_h} \cdot (T'_{A_v} - T'_{A_h})$$
$$\overline{T}_{B_h} = T'_{A_h} + \frac{\chi_h}{1 - \chi_v - \chi_h} \cdot (T'_{A_h} - T'_{A_v}) \,. \tag{A7}$$

For the computation of the SSM/I $T_B$s, a fixed set of APCs is used, taken from Wentz (1991) instead of sensor specific values. Colton and Poe (1999) concluded it would be best to use a fixed set of APCs for all sensors for a greater consistence across the sensors. The normalisation of the polarisation leakage factors as defined in Wentz (1991) $(\chi'_p)$ is different compared to the definition in Eq. (A3), but can be converted by:

$$\chi_p = \frac{\chi'_p}{1 + \chi'_p} \,. \tag{A8}$$

For the 22 GHz channel no vertical polarisation is measured and Wentz (1991) fitted a linear function with a least square regression to modelled oceanic $T_B$s and found

$$\overline{T}_{B_{v22}} = 1.01993 \cdot T_{A_{v22}} + 1.994 \,. \tag{A9}$$

The uncertainty of the approximation is about 0.1 K. Another function was fitted for land observations. However, the differences to the expression over ocean are smaller than 0.2 K and only the equation above is applied.

**A2  SSMIS**

The SSMIS has six individual feedhorns, which are placed at fixed angles from about -6° to 6° along the focal plane of the reflector. This alignment leads to an additional polarisation mixing, which has to be accounted for. The radiation entering each individual antenna feedhorn $T_{A^*_{v,h}}$ is therefore a composition of the polarised radiation depending on the effective polarisation angle $\gamma$ (similar to Eq. (19)):

$$T_{A^*_p} = T_{A_p} \cdot \cos^2 \gamma + T_{A_q} \cdot \sin^2 \gamma \,. \tag{A10}$$

This set of equations can be solved to derive the antenna temperature aligned with the Earth polarisation vector:

$$T_{A_p} = \frac{T_{A^*_p} \cdot \cos^2 \gamma - T_{A^*_q} \cdot \sin^2 \gamma}{\cos^2 \gamma - \sin^2 \gamma} \,. \tag{A11}$$

This rotation correction is applied to the SSM/I like channels at 19, 37, and 91 GHz before the antenna spillover Eq. (21) and cross-polarisation Eq. (A7) are corrected similar to the SSM/I. For the sounding channels only a spillover correction is applied.

As for the SSM/I, there is no horizontal polarisation measured at 22 GHz. In this case a synthetic $T_B$ is estimated from the $T_B$s at 19 GHz and 22 GHz from a least square regression:

$$\overline{T}_{B_{h22}} = c_1 + c_2 \overline{T}_{B_{h19}} + c_3 \overline{T}_{B_{v19}} + c_4 \overline{T}_{B_{v22}} \,. \tag{A12}$$





The values for coefficients $c$ are taken from the operational ground processing software configuration. Substituting this estimate into Eq. (A6) while accounting for the polarisation rotation correction Eq. (A10) and defining:

$$\beta_1 = (1 - \chi_v) \cos^2 \gamma + \chi_v \sin^2 \gamma$$
$$\beta_2 = (1 - \chi_v) \sin^2 \gamma + \chi_v \cos^2 \gamma \qquad (A13)$$

yields:

$$\overline{T}_{B_{v22}} = \frac{T'_{A_{v22}} - \beta_2 \cdot (c_1 + c_2 \overline{T}_{B_{h19}} + c_3 \overline{T}_{B_{v19}})}{\beta_1 + c_4 \beta_2} \ . \qquad (A14)$$

## A3   SMMR

Additional to the expected internal cross-polarisation leakage at the instruments polarisation selector switches, large polarisation mixing is induced as a result of the instruments design. The antenna system consists of a scanning parabolic reflector and a fixed multispectral feedhorn. Due to the antenna oscillating rotation, the alignment of the antenna pattern polarisation vector relative to the Earth surface changes with increasing scan angle $\phi$. With increasing scan angle, vertical polarisation is mixed into horizontal polarisation and vice versa. Several studies (Gloersen et al., 1980; Gloersen, 1987; Francis, 1987) addressed

this problem. The procedure used in the level 1B data records to correct for this effect was developed in Gloersen (1987) and implemented as follows:

The scan angle dependent integration of the antenna pattern over the antenna FOV can be derived similar to Eq. (A6) while keeping the antenna polarisation vector dependence of the antenna gains:

$$T'_{Ap}(\phi) = (1 - \chi_p(\phi)) \cdot \overline{T}_{Bp} + \chi_p(\phi) \cdot \overline{T}_{Bq} \qquad (A15)$$

The antenna pattern cross-polarisation leakage and switch polarisation mixing effects are combined and approximated as follows (Njoku et al., 1998):

$$\chi_p(\phi) = Q_p \cdot \sin^2(\phi - \delta_p) \ , \qquad (A16)$$

where $Q_p$ and $\delta_p$ are the amplitudes and phase shifts of the polarisation mixing. The final set of coefficients is estimated empirically from in-orbit data and given in Njoku et al. (1998). Eq. (A15) is then inverted to compute the scene $T_B$s from the

normalised antenna temperatures:

$$\overline{T}_{B_p} = T'_{A_p} + \frac{\chi_p(\phi)}{1 - \chi_q(\phi) - \chi_p(\phi)} \cdot (T'_{A_p} - T'_{A_q}) \ . \qquad (A17)$$

*Author contributions.*   All authors contributed to the generation of the data record and to the writing of the paper.

*Competing interests.*   The authors declare that they have no conflict of interest.



*Acknowledgements.* This work was performed within the EUMETSAT CM SAF framework and we acknowledge the financial support of
the EUMETSAT member states. The Nimbus-7 SMMR Pathfinder Level 1B Brightness Temperature data record was provided by the DAAC
at the National Snow and Ice Data Center. The NOAA Climate Data Record of SSM/I and SSMIS Microwave Brightness Temperatures was
provided by the National Centers for Environmental Information and the Colorado State University.



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

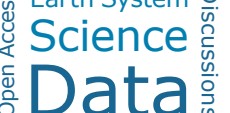

**Table 1.** Summary of estimated systematic uncertainty source contributions.

| Systematic uncertainty source | Systematic uncertainty range $\Delta a$ [K] | Standard uncertainty $u$ [K] |
|---|---|---|
| Warm load reference | - | 0.10 |
| Cosmic background reference | - | 0.10 |
| Calibration non-linearity SSMI(S) | 0.25 - 0.70 | 0.15 - 0.40 |
| Calibration non-linearity SMMR | - | 0.10 |
| Radiative coupling | 0.10 - 0.40 | 0.06 - 0.25 |
| Cross polarisation | 0.15 - 0.30 | 0.10 - 0.20 |
| Feedhorn spillover | 1.00 - 1.50 | 0.60 - 0.90 |
| Reflector emissivity SSMIS | 0.60 | 0.40 |
| Polarisation mixing SMMR | 0.20 - 0.50 | 0.10 - 0.30 |
| Total SMMR | | 0.40 - 1.00 |
| Total SSMIS | | 0.70 - 1.10 |

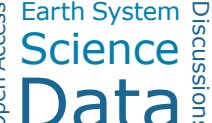

**Raw Data Record**

**Input/Output**     **Process**

- Decode raw data record
- Quality control and filtering
- Revert Earth counts (SSM/I, SSMIS)
- Geolocation
- Land sea type classification
- First guess radiometric calibration
- Reformat to common data format

**Land Mask**

**Level-1a data record**     **Detect intrusions**

- Final radiometric calibration
- Application of sensor specific corrections
- Computation of brightness temperatures
- Computation of intercalibration offsets

**Intrusion lookup maps**

**Level-1b data record**     **Derive daily sea ice masks**

- Final surface type classification
- Computation of EIA normalisation offsets
- Antenna pattern matching
- Final quality control

**Daily sea ice masks**

**FCDR**

**Figure 1.** Generalised data flow diagram with main processing steps to generate the CM SAF FCDR.



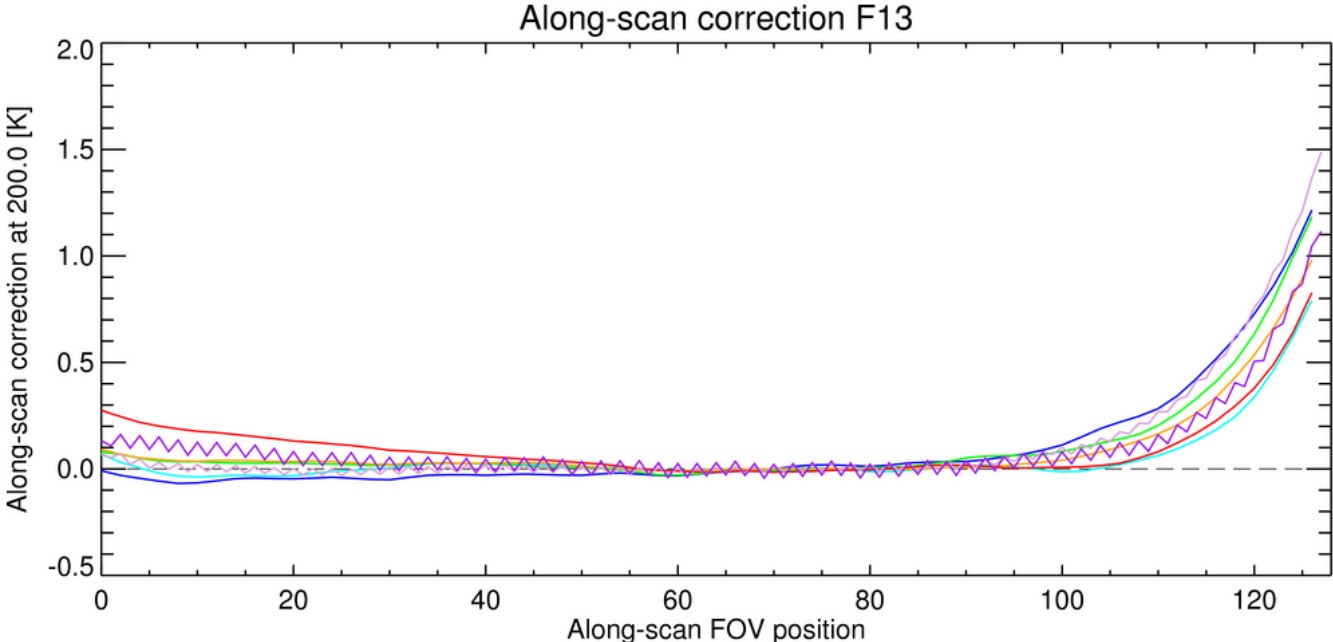

**Figure 2.** Along-scan correction for SSM/I$_{F13}$. The image shows the along-scan correction applied to a scene temperature of 200 K. Colours are as follows: cyan v19, blue h19, green v22, orange v37, red h37, plum v85, violet h85.

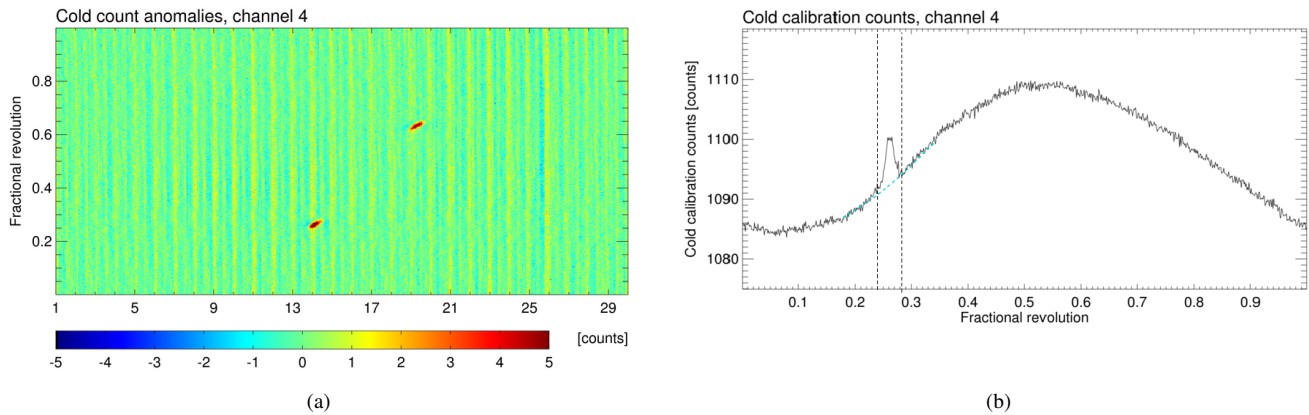

**Figure 3.** Cold count anomaly (a) from SSM/I$_{F13}$ channel at 37 GHz,v-pol for November 2005. The y-axis represents the fractional orbit angle with ascending equator crossing at 0 and descending equator crossing at 0.5. The x-axis represents the time of the orbit start at ascending equator crossing. A cross-section of the absolute cold counts through the maximum of the first intrusion event is shown on the right (b). Positions with identified cold count anomalies are between the vertical lines. The locally fitted reconstruction spline is plotted in cyan.



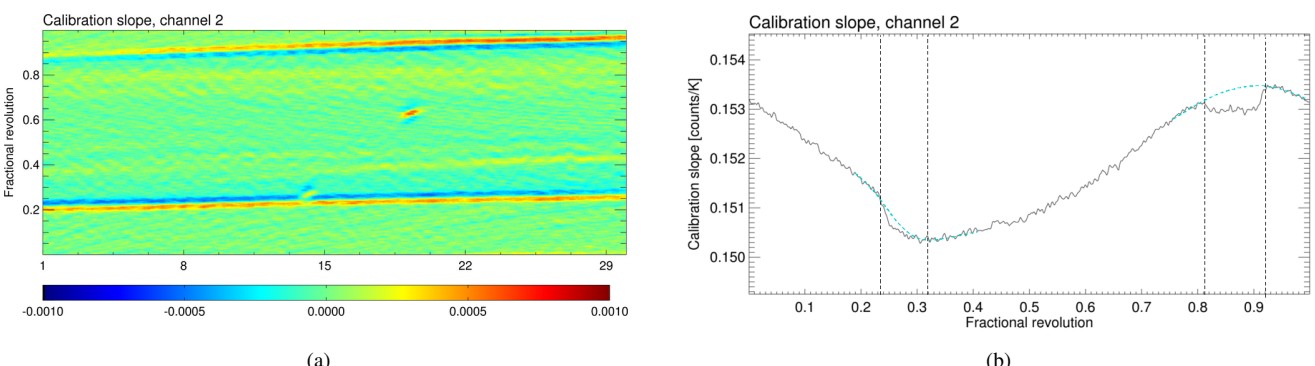

(a)

**Figure 4.** Edge detection results (Laplace operator) for calculated first guess slopes from SSM/I$_{F13}$ channel at 19 GHz,h-pol for November 2005 (a) and a cross-section of the first guess slope (b) with two sunlight intrusion events. The satellite leaves the Earths shadow approximately at position 0.28 and enters the Earths shadow at position 0.85. Positions with identified warm target anomalies are between the vertical lines. The locally fitted reconstruction splines are plotted in cyan.

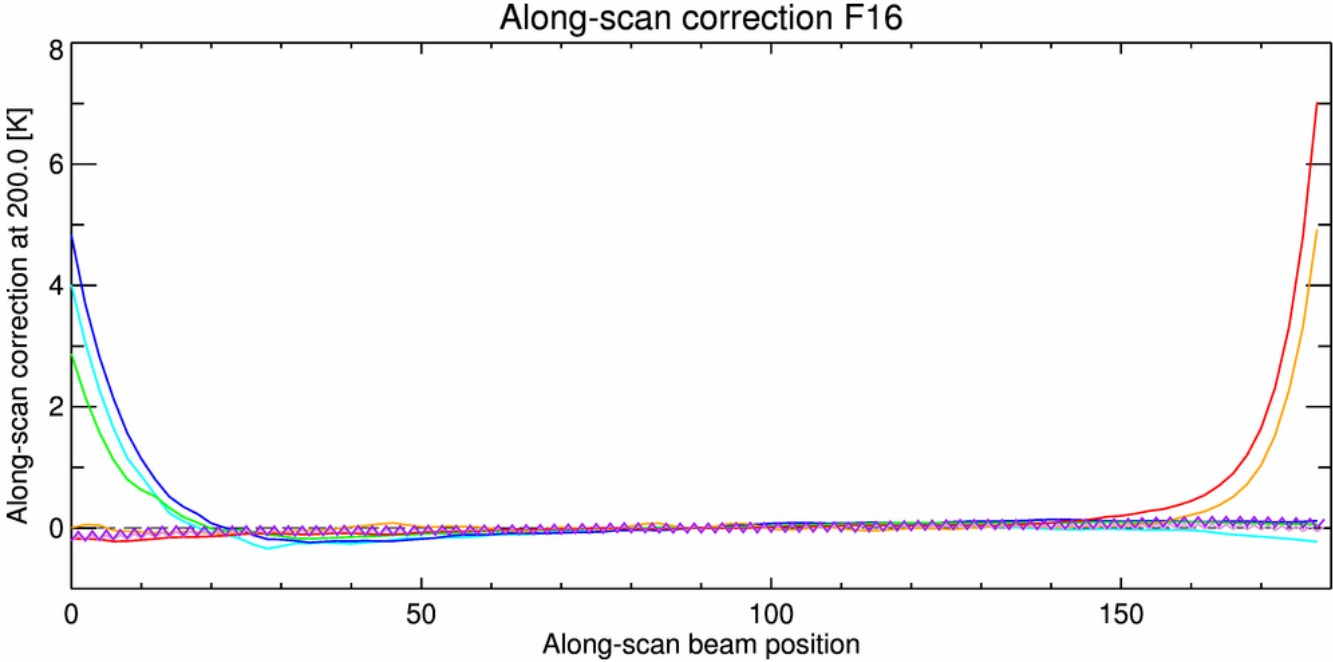

**Figure 5.** Along-scan correction for SSMIS$_{F16}$. The image shows the along-scan correction applied to a scene temperature of 200 K. Colours are as follows: cyan v19, blue h19, green v22, orange v37, red h37, plum v91, violet h91.



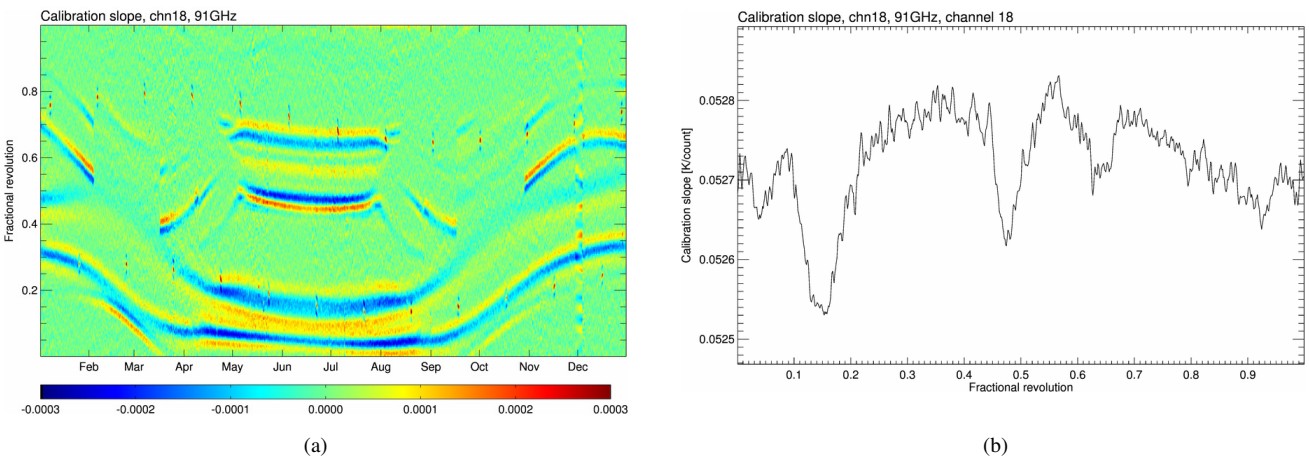

(a)                                                                                     (b)

**Figure 6.** Edge detection results (Laplace operator) for calculated first guess slopes from SSMIS$_{F16}$ channel at 91 GHz,h-pol for 2010 (a) and a cross-section of first guess slope (b) for one orbit in June 2010 (right). Four regions with warm load intrusion are identified around orbit positions 0.04, 0.15, 0.48, and 0.65.

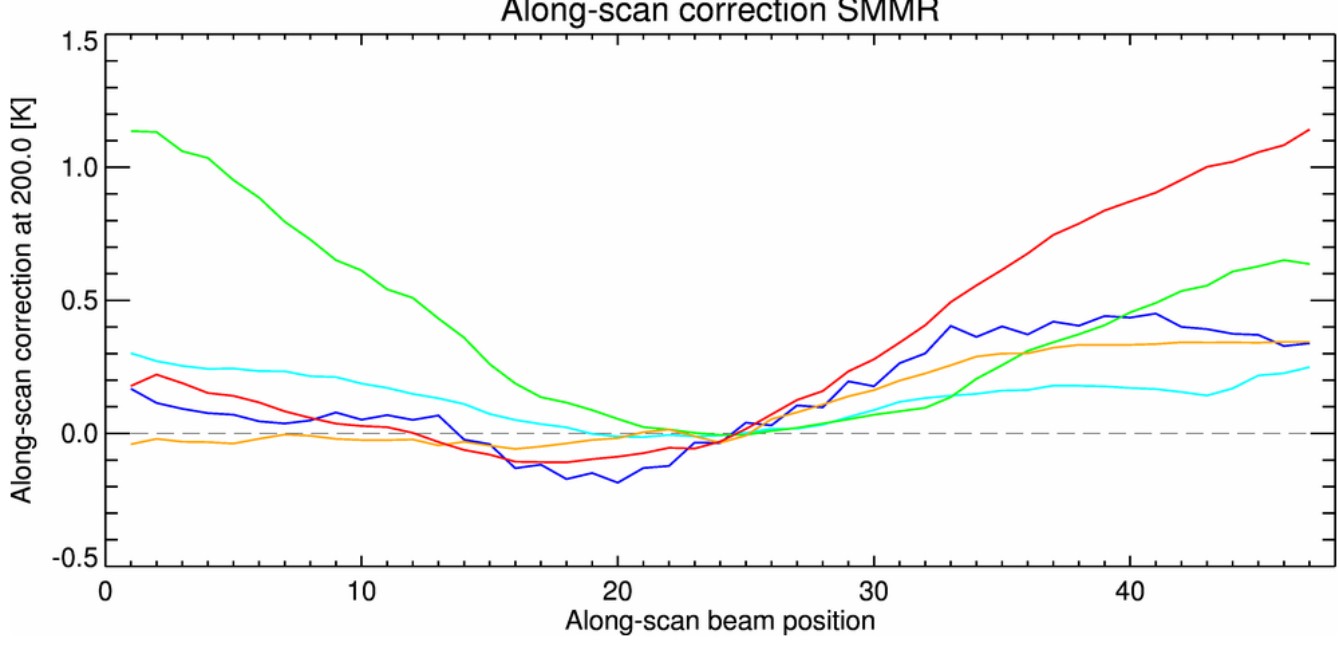

**Figure 7.** Scan angle dependent correction for the SMMR instruments. This image shows the along-scan correction at a common scene temperature of 200 K. Nadir is at position 24. Channel colours are as follows: cyan h18, blue v18, green v21, orange h37, red v37.






**Figure 8.** Time series of ensemble anomalies for SSM/I and SSMIS channel at 37 GHz,h-pol. In the upper two panels the solid lines are PM orbits and the dashed lines AM orbits. The lower panel shows daily means of AM and PM orbits. The grey lines depict the ensemble spread. For a detailed description see text. Colours are as follows: F16 orange, F17 blue, F18 black, F13 green and F14 purple.

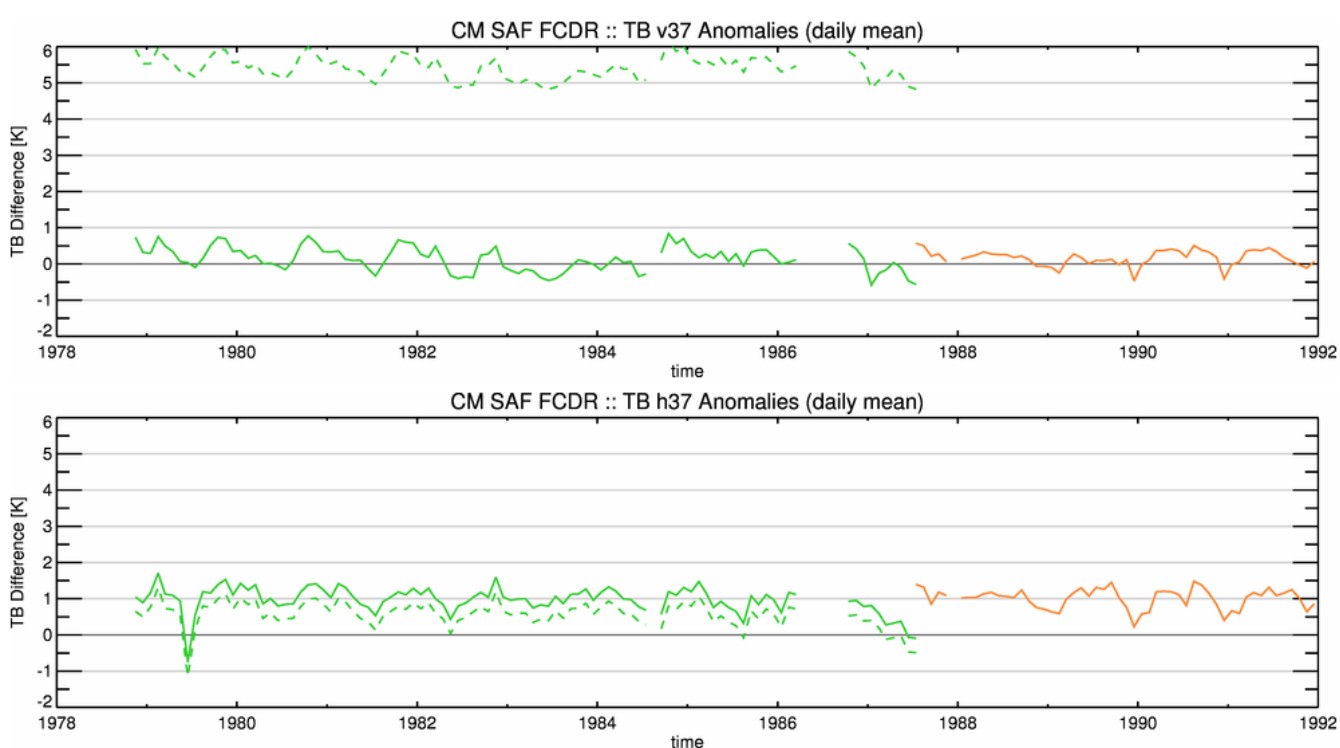

**Figure 9.** Time series of global monthly mean differences between observed and modelled $T_B$s at 37 GHz,v-pol (top) and 37 GHz,h-pol (bottom) before (dashed lines) and after (solid lines) inter-calibration of SMMR (SMMR, green; F08 orange).



**Figure 10.** Time series of global monthly mean $T_B$ differences for the 37 GHz,h-pol channel between the CM SAF FCDR and ERA-20C without inter-calibration (top), with inter-calibration (middle) and the $RSD$ of the inter-calibrated data record and ERA-20C (bottom). Colours for the instruments are SMMR (green), F08 (orange), F10 (blue), F11 (black), F13 (green), F14 (violet), F15 (red), F16 (orange), F17 (blue), F18 (black).