# Peer review of "A Fundamental climate data record of SMMR, SSM/I, and SSMIS brightness temperatures"

_Earth System Science Data, 2019_

## Referee Comment (RC1) · Thomas Lavergne (Referee) · 3 Oct 2019

**Review of essd-2019-146:** *A Fundamental climate data record of SMMR, SSM/I, and SSMIS brightness temperatures* **by Fennig et al.**

This paper introduces an homogeneized and inter-calibrated data record of conically scanning passive microwave radiometer brightness temperature data covering 1978 – 2015. The paper provides a thorough description of the various steps involved, an analysis of the random and systematic uncertainties, and some results of an evaluation. The dataset itself is already made available by the CM SAF, along with ample documentation (ATBD, PUM, Validation Report). Although most of the material in the paper is extracted from the documentation, the paper is usefull as a one-stop-shop reference.

The manuscript is very clear and comprehensive, and -as a user of the data record- I can only praise the attention to details and thoroughness of the paper. I highly recommend the publication of this manuscript in ESSD.

I include below some comments that will hopefully be usefull for improving the manuscript further.

**High-level comment:**
The manuscript explains very well how the data record is prepared (ATBD), give some example of evaluation results (Validation Report) but does not touch into how to use the product (PUM). A PUM exists so this is not critical, but I would still encourage the authors to add a short section describing what the data record consists of (e.g. daily aggregated files per sensor, use of groups in the netCDF files, quality flags, etc...).

Along those lines, as a user of the dataset, it took me some time to understand that the data read from the netCDF files was the homogeneized (but NOT intercalibrated) data, and that an extra step was required to build the intercalibrated record. You have in this paper the opportunity to clearly (re-)state what the data files contain, and how to use them (adding the various correction layers). This terminology (homogenised, inter-calibrated) is anyway needed to understand the evaluation results, e.g. in Figure 8. Maybe the same terms can appear on Figure 1?

**Detailed minor comments:**
Line 35: This is the 1st time you introduce CM SAF. Define the acronym, or refer to the FCDR differently.

Line 60: Suggest to rephrase ("This is not the only FCDR of passive microwave radiometry...").

Line 61: remove "respective"

Line 66: "This" data record… The GPM one, or the CM SAF one?

Line 72: please finalize your introduction with: 1) a discussion as to why the CM SAF FCDR was at all needed when others exist (e.g. wasn't the Wentz FCDR non-traceable and not free?) and 2) a short introduction to the structure of your paper.

Line 89: the high-frequency channels of F08 went bad quickly, maybe this can be noted here?

Line 90: Please note that the "footprint" is an instantaneous field-of-view. Also that you refer to the diameters of the 3dB ellipses. Cross- and along-track terminology could be introduced.

Line 106: F19 being launch in 2014 (before the end of the FCDR), your statement in the Abstract ("all SSM/I and SSMIS instruments") is not strictly correct. But this is probably ok.

Line 197: Suggest to refer here to section 3.4 for further details on the sea-ice concentration and masking.

Line 283: Could you add a reference/citation for the 13 leap seconds?

Line 285: "It can take up to 7 days before a leap second is introduced to the data record" is this in the original RDR? Also, at how many occasions (out of 13) was the leap second introduced inconsistently between observation and ephemeris? If not many, mention the years?

Line 314: Consider changing heading to "Antenna pattern matching for high-frequency channels".

Line 315: Change "it is important" to something like "it can be desirable"... many data producers choose to retain the high(er) resolution of the channels, at the cost of increase retrieval uncertainties.

Line 320: remove comma after "both"

Line 557: the «fence is working well»: add «(not shown here)» in this sentence already.

Line 674: Suggest to chagne header to «Intercalibration of sensors»

Line 683: please add a citation/reference for the F11 wind retrieval stability results.

Line 698: «highly elliptical orbit» (HEO) is often used for a spacecraft with Molniya orbit. Re-formulate to «orbit with higher ellipticity» ?

Line 716: Use the exact frequencies for the instruments. Do we really want to mention 85 GHz here (SSMIS had 91 Ghz)? Re-formulate so that to make clear what you do with the high-frequency channels.

Eq 25: Can you name what the «#» Ts are? Should you add a sentence below this equation to re-name the various terms? For example $<T^h>$ is defined in Eq 6 which was quite some pages ago.

Eq 26: I could not find other occurences of «APC» used as an operator in the text, and am unsure what APC(T#A) means in practice. Does $T^{ic}$ means «inter-calibrated»? By adding text around Eq 25 and 26 you will help the reader.

L736: «on the lower SSMI(S) resolutions» do you mean «frequencies» here?

L742: Should the two sentences: «This selection of TBs uses the Earth ... thus no extrapolation is required.» be moved at the end of the paragraph? Currently they seem to fall in the middle of the description of your matchup-database (first described as morning/afternoon maps, then to monthly).

L744: would it be more correct to say that «little interpolation is needed»? One could imagine that the averaging in monthly 1x1deg grids will damper extremes that will be (slightly) outside the vicarious calibration range...

L751: so if I understand correctly, the ocean scenes entering the inter-calibration (cold vicarious target) use the angle correction, but not the sea-ice or land (hot vicarious target). I suggest you add a sentence to make this explicit to the reader, and maybe discuss why this is a viable approach.

L781: So contrarily to SSMI(S) only a cold vicarious calibration target is used, correct?

L1046: where does the range 0.7 K to 1.1 K comes from? Add a citation, or a cross-reference to one of your sections.

L1037: «EIA normalisation and diurnal cycle». I get what the EIA normalization is, but what is the correction for diurnal cycle? Where is it described?

L1038 to L1064: a suggestion is to refer to Figure(s) from the ATBD when referring to «not shown» results.

L1065: suggestion to rename this heading to something like «double differences for SMMR» (I am aware SSM/I$_{F08}$ results are shown, but the core is SMMR?).

L1156: This is commented here, but should probably be addressed at an early stage in the manuscript. What do you mean with RDR exactly? Is it the SSM/I and SSMIS data you first accessed as a source to building your FCDR? Can you refer to it with a

L1171: ... and Lavergne et al. (2019) used the full FCDR for building their sea-ice concentration data record.

L1175: Since no AMSRs are currently on-board you could refer to the family of AMSRs rather than the specific AMSR-3 (which is not firmly commited at time of writing). An FCDR of AMSRs compatible with the SMMR+SSM/I+SSMIS CMSAF FCDR would be greatly beneficial for many applications, including sea-ice.

L1175: You spent a lot of efforts (and text) due to accessing Njoku's SMMR L1B (instead of $T_a$s) which was a limitation to your harmonization process. Are you aware of plans for somoeone to release the «raw» SMMR data record, so that you could improve the first part of your FCDR? I would have added here some sentences calling for such a release, especially if (funded) data rescue activities must be activited.

L1250: you could add an acknowledgement for ERA20C.

Figure 2. Because some colors are rather similar (e.g. plum and violet) it would help if a legend box was added in the plot area. Consider using thicker lines.

Figure 2. What causes the up-and-down variations for some channels (seemingly the high-frequency ones). Is it because the along-scan correction is different for A- and B-scans? If the case, would it be better to show 2 lines per high-frequency channel? If needed, add a sentence L474 about this feature.

Figure 3, left panel. «The x-axis represents the time of the orbit start at ascending equator crossing». Is the x-axis with unit «day»? If so, add it. Did you consider using a red-gray-blue colormap to avoid the rainbow one? On right panel, thicker lines would help.

Figure 4, same remarks as Figure 3.

Figure 5, same remarks as Figure 2.

Figure 6, same remarks as Figure 3. In panel 6b you do not show sections that are detected by the Laplace filter, nor the smoothed spline. Is it intentional?

Figure 7, same remarks as Figure 2. According to the caption, you use cyan for h18 (horizontal pol) while you were using it for v19 (vertical pol) for SSM/I (Fig 2) and SSMIS (Fig 5). Is this intentional? It would be better to use the same colors for all sensors.

Figure 8: add a legend box for the line colors. I do not understand what the grey lines are. Do they show a spread value (1-standard deviation?) between the available sensors within one day, while the colored lines are the mean daily anomalies? Please clarify.

Figure 9 and 10: add a legend box for the line colors.

---

## Referee Comment (RC2) · Catherine Prigent (Referee) · 17 Oct 2019

General comments:

This paper presents a detailed and careful analysis of the original observations from SMMR, SSMI and SSMIS, to produce a high quality Fundamental Climate Data Record of passive microwave brightness temperatures. It summarizes a long term effort from the Climate SAF group. This FCDR is widely used by the passive microwave community, for multiple applications including reanalysis exercises in NWP centers. The document is a very informative and well written description of the different steps needed to obtain the FCDR, with a clear and honest quantification of the errors. This paper has to be rapidly published, after minor corrections.

[Figure]

Our group uses this FCDR extensively. We appreciate the quality of the data, as well as the responsiveness of the Climate SAF to answer any question related to the dataset.

Minor comments:

- Sections 3.6.3 and 3.6.6. These subsections might be too detailed. The figures include a lot of information that is not fully explained (e.g., axis, color scales). It should be possible to improve these figures. The orbit position is mentioned several times in the text. In the legend of the figures it is called fractional revolution. Can you clarify?

- P 23. DMSP F11 is used as the reference, with different reasons to justify this choice. However, it might be worth mentioning that the overpassing time of this instrument drifted significantly during its life time, with large time differences with F08, F10, F13, and F14 during their overlapping periods. That can have potential effects on the inter-calibration, especially over land.

- P 25. L 740. The warm surface types are only considered for their polarization differences. Checking the inter-calibration of the polarization differences over warm scene is certainly informative, but how can it make sure that the TbV and the TbH are independently correctly inter-calibrated from an instrument to the next? The warm and stable targets are usually selected over the Amazon forest that shows a very small polarization difference (both TbV and TbH warm). Over deserts, it would be possible to have rather high polarization difference with TbV high and consequently TbH rather low, but over deserts, the diurnal variation of the surface temperature is large and would make it very difficult to compare instruments that do not have the same overpassing times. As a consequence, it is difficult to understand how the inter-calibration takes into account the full temperature range, including the warm scenes. Can the authors elaborate on this point?

- P 37. L 1118. Uncertainty of the radiative transfer model due to scattering effects. . . Scattering can play a role, but limited for frequencies below ∼50 GHz. Uncertainties are more likely due to the lack of realistic cloud and rain information to feed the radiative

transfer model, for all cloud and rain effects (emission, attenuation, and scattering).

- P 38. L 1149 1150. FCDR... includes all possible surface types... Would it be relevant to mention that over land the inter-calibration might be less robust than over ocean, given that some procedures are only applicable over ocean (L 712-714), and some others are only taking into account part of the warm scene signal (L 740)? A word of caution for the users of the FCDR over land could be helpful.

- Figures 2 to 10 would certainly benefit from some additional work. The axes should be clarified, with mention of the units. The legend of the different line colors should be added to the figures.

Technical corrections:

- P 5. The spatial resolution of the SMMR instrument is not mentioned, whereas this information is provided for the other instruments

- P 6. L 181. MD5 hash: can you provide a reference and / or a few words of explanation?

- P 9. L 268-270. Channel numbers are not used elsewhere. Better mention their frequencies?

- P 13. L 387. ... each scan pair that passES the quality control...

- P 13. L 400. ... Earth'S counts

- P 26. L 798. ... trends cloud have been not accounted for... Rephrase?

- P 31. L 933. The on-orbit calibration IS...

- P 32. L 972. Instrument design (suppress the S).

- P 33. L 1009. A factor of 1.48. Where does this factor come from?

- P 35. L 1044. This means that most....

- P 35. L 1052. . . . and showS very similar. . .

- P 35. L 1057. This variability is caused by. . . Which variability are the authors talking about? The increase variability with frequency or the fact that the H polarization variability is larger than the vertical one? Rephrase to clarify?

- P 37. L 1112. It was further shown that. . . suppress the coma.

- P 37. L 1133. It becomes clear that. . . suppress the coma.
* * *

---

## Author Comment (AC1) · 11 Dec 2019

**Catherine Prigent (Referee #2)**

General comments:

This paper presents a detailed and careful analysis of the original observations from SMMR, SSMI and SSMIS, to produce a high quality Fundamental Climate Data Record of passive microwave brightness temperatures. It summarizes a long term effort from the Climate SAF group. This FCDR is widely used by the passive microwave community, for multiple applications including reanalysis exercises in NWP centers. The document is a very informative and well written description of the different steps needed to obtain the FCDR, with a clear and honest quantification of the errors. This paper has to be rapidly published, after minor corrections. Our group uses this FCDR extensively. We appreciate the quality of the data, as well as the responsiveness of the Climate SAF to answer any question related to the dataset.

Minor comments:

*Referee comment*

Sections 3.6.3 and 3.6.6. These subsections might be too detailed. The figures include a lot of information that is not fully explained (e.g., axis, color scales). It should be possible to improve these figures. The orbit position is mentioned several times in the text. In the legend of the figures it is called fractional revolution. Can you clarify?

*Author's response*

The orbit position in the figures 3, 4, and 6 is presented as fractional revolutions, measured from the start of the orbit, which is when the spacecraft crosses the equator from south to north until one revolution is finished. In the figures 3a, 4a, and 6a each image column is exactly one orbit with the start of the orbit as the x-axis and the fractional revolution (from one equator crossing to the next) on the y-axis. This is also mentioned in the caption of figure 3 where it is first used.

*Author's changes to the manuscript*

We will update the figures to add axis labelling and colour information where it is missing. See also RC1 comments regarding figures. We will also explain the term fractional revolution in section 3.6.2 where figure 3 is referenced.

*Referee comment*

P 23. DMSP F11 is used as the reference, with different reasons to justify this choice. However, it might be worth mentioning that the overpassing time of this instrument drifted significantly during its life time, with large time differences with F08, F10, F13, and F14 during their overlapping periods. That can have potential effects on the intercalibration, especially over land.

*Author's response*

Yes, there is a drift in the F11 overpass time. The drifting of all DMSP satellites is a general issue for the inter-calibration. We are minimizing the impact of the diurnal cycle in the homogenisation step of the inter-calibration.

*Author's changes to the manuscript*

We will mention the overpass drifting of the satellites and note the potential effects on the inter-calibration in section 5.

*Referee comment*

P 25. L 740. The warm surface types are only considered for their polarization differences. Checking the inter-calibration of the polarization differences over warm scene is certainly informative, but how can it make sure that the TbV and the TbH are independently correctly inter-calibrated from an instrument

to the next? The warm and stable targets are usually selected over the Amazon forest that shows a very small polarization difference (both TbV and TbH warm). Over deserts, it would be possible to have rather high polarization difference with TbV high and consequently TbH rather low, but over deserts, the diurnal variation of the surface temperature is large and would make it very difficult to compare instruments that do not have the same overpassing times. As a consequence, it is difficult to understand how the inter-calibration takes into account the full temperature range, including the warm scenes. Can the authors elaborate on this point?

*Author's response*

The point raised here is actually the reason why the TB polarisation difference is used explicitly in the inter-calibration. The polarisation difference over land must be preserved during the inter-calibration. The inter-calibration offset and scale factors of TBh and TBv are therefore not independent. The TB polarisation difference term, which is used for all surface types, acts as a constraint to the scaling factor. As a consequence, the inter-calibration scaling factors and offsets are mainly determined over the ocean at the cold end of the natural variability but are constrained over the whole spectrum (cold ocean, warm land).

*Author's changes to the manuscript*

We will extend the explanation of the inter-calibration model in section 5.2.

*Referee comment*

P 37. L 1118. Uncertainty of the radiative transfer model due to scattering effects: : : Scattering can play a role, but limited for frequencies below ~50 GHz. Uncertainties are more likely due to the lack of realistic cloud and rain information to feed the radiative transfer model, for all cloud and rain effects (emission, attenuation, and scattering).

*Author's response*

We agree that the unknown atmospheric state to feed the RTM has a significant impact. From our experience with SSM/I data, a scattering effect is observable for 37GHz and higher frequencies.

*Author's changes to the manuscript*

We will add the unknown atmospheric state to the uncertainty description in section 7.3.

*Referee comment*

P 38. L 1149 1150. FCDR: : : includes all possible surface types: : : Would it be relevant to mention that over land the inter-calibration might be less robust than over ocean, given that some procedures are only applicable over ocean (L 712-714), and some others are only taking into account part of the warm scene signal (L 740)? A word of caution for the users of the FCDR over land could be helpful.

*Author's response*

Yes, you are right. We have mentioned this limitation in the documentation of the FCDR but missed to mention it in the paper.

*Author's changes to the manuscript*

We will add the information in section 8 to explain a higher uncertainty of the inter-calibration over land areas.

*Referee comment*

Figures 2 to 10 would certainly benefit from some additional work. The axes should be clarified, with mention of the units. The legend of the different line colors should be added to the figures.

*Author's response*

Yes, the figures 2 – 10 can be improved.

*Author's changes to the manuscript*

We will improve the figures by adding colour bars, units and axis legends where missing.

*Referee comment*

Technical corrections:
- P 5. The spatial resolution of the SMMR instrument is not mentioned, whereas this information is provided for the other instruments

- P 6. L 181. MD5 hash: can you provide a reference and / or a few words of explanation?

- P 9. L 268-270. Channel numbers are not used elsewhere. Better mention their frequencies?

- P 13. L 387. : : : each scan pair that passES the quality control: : :

- P 13. L 400. : : : Earth'S counts

- P 26. L 798. : : : trends cloud have been not accounted for: : : Rephrase?

- P 31. L 933. The on-orbit calibration IS: : :

- P 32. L 972. Instrument design (suppress the S).

- P 33. L 1009. A factor of 1.48. Where does this factor come from?

- P 35. L 1044. This means that most: : :.

- P 35. L 1052. : : : and showS very similar: : :

- P 35. L 1057. This variability is caused by: : : Which variability are the authors talking about? The increase variability with frequency or the fact that the H polarization variability is larger than the vertical one? Rephrase to clarify?

- P 37. L 1112. It was further shown that: : : suppress the coma.

- P 37. L 1133. It becomes clear that: : : suppress the coma.

*Author's response*

Thanks for the corrections.

*Author's changes to the manuscript*

We will modify the paper accordingly and correct/clarify these minor issues.

---

## Author Comment (AC2) · 12 Dec 2019

**Thomas Lavergne (Referee #1)**

**Review of essd-2019-146: A Fundamental climate data record of SMMR, SSM/I, and SSMIS brightness temperatures by Fennig et al.**

This paper introduces an homogeneized and inter-calibrated data record of conically scanning passive microwave radiometer brightness temperature data covering 1978 – 2015. The paper provides a thorough description of the various steps involved, an analysis of the random and systematic uncertainties, and some results of an evaluation. The dataset itself is already made available by the CM SAF, along with ample documentation (ATBD, PUM, Validation Report). Although most of the material in the paper is extracted from the documentation, the paper is usefull as a one-stop-shop reference.

The manuscript is very clear and comprehensive, and -as a user of the data record- I can only praise the attention to details and thoroughness of the paper. I highly recommend the publication of this manuscript in ESSD.

I include below some comments that will hopefully be usefull for improving the manuscript further.

*High-level comment:*
The manuscript explains very well how the data record is prepared (ATBD), give some example of evaluation results (Validation Report) but does not touch into how to use the product (PUM). A PUM exists so this is not critical, but I would still encourage the authors to add a short section describing what the data record consists of (e.g. daily aggregated files per sensor, use of groups in the netCDF files, quality flags, etc...). Along those lines, as a user of the dataset, it took me some time to understand that the data read from the netCDF files was the homogeneized (but NOT intercalibrated) data, and that an extra step was required to build the intercalibrated record. You have in this paper the opportunity to clearly (re-)state what the data files contain, and how to use them (adding the various correction layers). This terminology (homogenised, inter-calibrated) is anyway needed to understand the evaluation results, e.g. in Figure 8. Maybe the same terms can appear on Figure 1?

*Author's response*
We had not included these details yet but referenced the PUM for further information. We can add more details about the data record. We want to avoid adding more terms in Figure 1, but explain the terms in the text.

*Author's changes to the manuscript*
We include more detailed information about the data file content and usage in section 3. The definition of the used terminology is added where Figure 1 is explained (also in section 3).

**Detailed minor comments:**

*Referee comment*
Line 35: This is the 1st time you introduce CM SAF. Define the acronym, or refer to the FCDR differently.
*Author's response*
OK
*Author's changes to the manuscript*
We add a sentence explaining the role of CM SAF and define the acronym accordingly.

*Referee comment*
Line 60: Suggest to rephrase ("This is not the only FCDR of passive microwave radiometry...").
*Author's response*
OK
*Author's changes to the manuscript*
Changed accordingly.

*Referee comment*
Line 61: remove "respective"

*Author's response*
OK

*Author's changes to the manuscript*
Changed accordingly.

*Referee comment*
Line 66: "This" data record… The GPM one, or the CM SAF one?

*Author's response*
The GPM one is meant.

*Author's changes to the manuscript*
Clarified accordingly.

*Referee comment*
Line 72: please finalize your introduction with: 1) a discussion as to why the CM SAF FCDR was at all needed when others exist (e.g. wasn't the Wentz FCDR non-traceable and not free?) and 2) a short introduction to the structure of your paper.

*Author's response*
The structure of the paper is summarized from L67-L71. The FCDR was initiated because the Version 6 FCDR from RSS had deficiencies, was not available as TA and not documented/traceable. Also the FCDR from CSU was not available yet.

*Author's changes to the manuscript*
A short discussion for the need of the FCDR is added to introduction.

*Referee comment*
Line 89: the high-frequency channels of F08 went bad quickly, maybe this can be noted here?

*Author's response*
OK

*Author's changes to the manuscript*
Changed accordingly.

*Referee comment*
Line 90: Please note that the "footprint" is an instantaneous field-of-view. Also that you refer to the diameters of the 3dB ellipses. Cross- and along-track terminology could be introduced.

*Author's response*
OK

*Author's changes to the manuscript*
Changed/added accordingly.

*Referee comment*
Line 106: F19 being launch in 2014 (before the end of the FCDR), your statement in the Abstract ("all SSM/I and SSMIS instruments") is not strictly correct. But this is probably ok.

*Author's response*
Yes, strictly speaking you are right with "all", but we would like to keep the statement.

*Author's changes to the manuscript*
none

*Referee comment*
Line 197: Suggest to refer here to section 3.4 for further details on the sea-ice concentration and masking.

*Author's response*
OK, we missed this reference.

*Author's changes to the manuscript*
Add the reference.

*Referee comment*

Line 283: Could you add a reference/citation for the 13 leap seconds?

*Author's response*

The current list of leap seconds is available in Bulletin-C from this website:
https://www.iers.org/IERS/EN/Publications/Bulletins/bulletins.html

*Author's changes to the manuscript*

The reference to the IERS bulletin is added.

*Referee comment*

Line 285: "It can take up to 7 days before a leap second is introduced to the data record" is this in the original RDR? Also, at how many occasions (out of 13) was the leap second introduced inconsistently between observation and ephemeris? If not many, mention the years?

*Author's response*

Yes the leap second correction is done in the RDR, mostly several days after the official introduction of the leap second. This is quite complicated, because in some years the correction of ephemeris time and scan time is just a few seconds apart or for one sensor it is ok but not for the other one. It would be difficult to write a simple description without being too detailed. As we have corrected the times we think it is not necessary to list all the years.

*Author's changes to the manuscript*

None

*Referee comment*

Line 314: Consider changing heading to "Antenna pattern matching for high-frequency channels".

*Author's response*

OK

*Author's changes to the manuscript*

Heading changed accordingly.

*Referee comment*

Line 315: Change "it is important" to something like "it can be desirable"... many data producers choose to retain the high(er) resolution of the channels, at the cost of increase retrieval uncertainties.

*Author's response*

OK

*Author's changes to the manuscript*

Changed as suggested.

*Referee comment*

Line 320: remove comma after "both"

*Author's response*

OK

*Author's changes to the manuscript*

Changed accordingly.

*Referee comment*

Line 557: the «fence is working well»: add «(not shown here)» in this sentence already.

*Author's response*

OK

*Author's changes to the manuscript*

Changed accordingly.

*Referee comment*

Line 674: Suggest to chagne header to «Intercalibration of sensors»

*Author's response*

OK

*Author's changes to the manuscript*

Changed accordingly.

*Referee comment*
Line 683: please add a citation/reference for the F11 wind retrieval stability results.

*Author's response*
Add citation Andersson et al. (2010).

*Author's changes to the manuscript*
Add citation.

*Referee comment*
Line 698: «highly elliptical orbit» (HEO) is often used for a spacecraft with Molniya orbit. Reformulate to «orbit with higher ellipticity» ?

*Author's response*
OK

*Author's changes to the manuscript*
Sentence is reformulated to ".. due to an orbit with higher ellipticity."

*Referee comment*
Line 716: Use the exact frequencies for the instruments. Do we really want to mention 85 GHz here (SSMIS had 91 Ghz)? Re-formulate so that to make clear what you do with the high-frequency channels.

*Author's response*
Yes, the change from 85 to 91 GHz Channel is not a seamless data record.

*Author's changes to the manuscript*
Remove 91GHz from sentence and add: "The 85 GHz channels on the SSM/I are replaced with 91 GHz channels on the SSMIS. In order to allow the continued usage of existing algorithms, synthetic 85 GHz TBs are estimated from the 91 GHz TBs, inter-calibrated to the SSM/I time series and provided in the FCDR data files. Details can be found in the ATBD."

*Referee comment*
Eq 25: Can you name what the «#» Ts are? Should you add a sentence below this equation to re-name the various terms? For example <Th> is defined in Eq 6 which was quite some pages ago.

*Author's response*
These are the antenna/brightness temperatures with applied non-linear corrections.

*Author's changes to the manuscript*
Add a sentence to explain the terms.

*Referee comment*
Eq 26: I could not find other occurences of «APC» used as an operator in the text, and am unsure what APC(T#A) means in practice. Does Tic means «inter-calibrated»? By adding text around Eq 25 and 26 you will help the reader.

*Author's response*
The term APC is explained on L657 when the antenna pattern corrections are explained. Tic mean inter-calibrated.

*Author's changes to the manuscript*
Add a description of the terms and referencing the definition of the APC in section 4.

*Referee comment*
L736: «on the lower SSMI(S) resolutions» do you mean «frequencies» here?

*Author's response*
No, it means that we used the 91GHz channels after antenna pattern matching to the lower resolution, not the high resolution feedhorn for gridding.

*Author's changes to the manuscript*
Add a sentence for clarification.

*Referee comment*
> L742: Should the two sentences: «This selection of TBs uses the Earth ... thus no extrapolation is required.» be moved at the end of the paragraph? Currently they seem to fall in the middle of the description of your matchup-database (first described as morning/afternoon maps, then to monthly).

*Author's response*
> From our point of view It would be better to move the sentence "Each channel … orbits." behind these two sentences.

*Author's changes to the manuscript*
> Change sentence ordering.

*Referee comment*
> L744: would it be more correct to say that «little interpolation is needed»? One could imagine that the averaging in monthly 1x1deg grids will damper extremes that will be (slightly) outside the vicarious calibration range...

*Author's response*
> We agree that there is certainly some averaging of extreme values and there will be some minor scale extrapolation at these extreme ends. However, the intention here is to argue about a significant extrapolation over a range of 50K.

*Author's changes to the manuscript*
> Add a sentence: "Extrapolation is limited with respect to extreme values which are damped due to the averaging process."

*Referee comment*
> L751: so if I understand correctly, the ocean scenes entering the inter-calibration (cold vicarious target) use the angle correction, but not the sea-ice or land (hot vicarious target). I suggest you add a sentence to make this explicit to the reader, and maybe discuss why this is a viable approach.

*Author's response*
> Yes, just the ocean scenes are EIA normalised.

*Author's changes to the manuscript*
> Add a sentence for clarification that only ocean scenes are EIA normalised

*Referee comment*
> L781: So contrarily to SSMI(S) only a cold vicarious calibration target is used, correct?

*Author's response*
> Yes, that is correct. Here we follow Njoku (1998) as explained in L808-L810.

*Author's changes to the manuscript*
> None

*Referee comment*
> L1046: where does the range 0.7 K to 1.1 K comes from? Add a citation, or a crossreference to one of your sections.

*Author's response*
> These numbers are from the uncertainty analysis in section 6.

*Author's changes to the manuscript*
> Add a reference to section 6.

*Referee comment*
> L1037: «EIA normalisation and diurnal cycle». I get what the EIA normalization is, but what is the correction for diurnal cycle? Where is it described?

*Author's response*
> We have developed a method to fit a diurnal cycle correction for the SSM/I, which is described in the corresponding ATBD. However, as we are now using the daily mean gridded values to fit the inter-calibration coefficients, this method is not described in the paper. In the validation figures it is used to check the homogenisation for ascending/descending orbits.

*Author's changes to the manuscript*
> We add a sentence explaining the correction and referencing the ATBD for detailed information.

*Referee comment*

L1038 to L1064: a suggestion is to refer to Figure(s) from the ATBD when referring to «not shown» results.

*Author's response*

OK

*Author's changes to the manuscript*

Modifying the manuscript as suggested.

*Referee comment*

L1065: suggestion to rename this heading to something like «double differences for SMMR» (I am aware SSM/IF08 results are shown, but the core is SMMR?).

*Author's response*

OK

*Author's changes to the manuscript*

Modifying the manuscript as suggested.

*Referee comment*

L1156: This is commented here, but should probably be addressed at an early stage in the manuscript. What do you mean with RDR exactly? Is it the SSM/I and SSMIS data you first accessed as a source to building your FCDR? Can you refer to it with a

*Author's response*

Unfortunately, your sentence is not complete. The RDR are the raw data records as downloaded without any modification.

*Author's changes to the manuscript*

We add a description of RDR in the beginning of section 3 where the data processing is explained.

*Referee comment*

L1171: ... and Lavergne et al. (2019) used the full FCDR for building their sea-ice concentration data record.

*Author's response*

OK

*Author's changes to the manuscript*

Modify manuscripts as suggested.

*Referee comment*

L1175: Since no AMSRs are currently on-board you could refer to the family of AMSRs rather than the specific AMSR-3 (which is not firmly commited at time of writing). An FCDR of AMSRs compatible with the SMMR+SSM/I+SSMIS CMSAF FCDR would be greatly beneficial for many applications, including sea-ice.

*Author's response*

We are not planning to release a full FCDR for instruments when agencies have their own reprocessing activities. However, we are working on the inter-calibration of these sensors to homogenise these with our FCDR.

*Author's changes to the manuscript*

We will add a sentence describing our future plans with the AMSR series.

*Referee comment*

L1175: You spent a lot of efforts (and text) due to accessing Njoku's SMMR L1B (instead of Tas) which was a limitation to your harmonization process. Are you aware of plans for somoeone to release the «raw» SMMR data record, so that you could improve the first part of your FCDR? I would have added here some sentences calling for such a release, especially if (funded) data rescue activities must be activited.

*Author's response*

The RDR is still not available and we are not aware of any plans to recover these.

*Author's changes to the manuscript*

We add a sentence highlighting the need of the original data for an improved inter-calibration.

*Referee comment*
   L1250: you could add an acknowledgement for ERA20C.
*Author's response*
   OK
*Author's changes to the manuscript*
   Modify manuscripts as suggested.

*Referee comment*
   Figure 2. Because some colors are rather similar (e.g. plum and violet) it would help if a legend box was added in the plot area. Consider using thicker lines.
*Author's response*
   OK
*Author's changes to the manuscript*
   Modify accordingly and add a legend box.

*Referee comment*
   Figure 2. What causes the up-and-down variations for some channels (seemingly the highfrequency ones). Is it because the along-scan correction is different for A- and B-scans? If the case, would it be better to show 2 lines per high-frequency channel? If needed, add a sentence L474 about this feature.
*Author's response*
   From the technical description of the SSMIS we get the information that there always 2 sets of integrators for each channel, one for the odd and one for the even numbered positions. They are not perfectly matched and must be calibrated each scan. So the assumption is that the SSM/I 85GHz channels are similar and small differences between these two integrators remain.
*Author's changes to the manuscript*
   Add a sentence to offer an explanation of this behaviour.

*Referee comment*
   Figure 3, left panel. «The x-axis represents the time of the orbit start at ascending equator crossing». Is the x-axis with unit «day»? If so, add it. Did you consider using a red-gray-blue colormap to avoid the rainbow one? On right panel, thicker lines would help.
   Figure 4, same remarks as Figure 3.
   Figure 5, same remarks as Figure 2.
*Author's response*
   We will change the colour bar to avoid the rainbow colour bar.
*Author's changes to the manuscript*
   Change the colour bar and use thicker lines.

*Referee comment*
   Figure 6, same remarks as Figure 3. In panel 6b you do not show sections that are detected by the Laplace filter, nor the smoothed spline. Is it intentional?
*Author's response*
   Yes, this is intentional. Here we just want to show the different regions with sunlight intrusions not showing the correction, which is already done in the SSM/I figure.
*Author's changes to the manuscript*
   None

*Referee comment*
   Figure 7, same remarks as Figure 2. According to the caption, you use cyan for h18 (horizontal pol) while you were using it for v19 (vertical pol) for SSM/I (Fig 2) and SSMIS (Fig 5). Is this intentional? It would be better to use the same colors for all sensors.
*Author's response*
   OK
*Author's changes to the manuscript*
   We change the figure colours as suggested.

*Referee comment*
Figure 8: add a legend box for the line colors. I do not understand what the grey lines are. Do they show a spread value (1-standard deviation?) between the available sensors within one day, while the colored lines are the mean daily anomalies? Please clarify.

*Author's response*
The grey line are the maximum differences observed between any 2 sensors (sensor ensemble spread).

*Author's changes to the manuscript*
Clarify in manuscript.

*Referee comment*
Figure 9 and 10: add a legend box for the line colors.

*Author's response*
OK

*Author's changes to the manuscript*
Add legend box.

---

## Author Response (AR1)

[revised manuscript text omitted]
                       |                                             | 0.70 - 1.10                  |